# Lanthanide luminescence nanothermometer with working wavelength beyond 1500 nm for cerebrovascular temperature imaging in vivo

Yukai Wu [1,5], Fang Li[1,5], Yanan Wu [2], Hao Wang[1], Liangtao Gu[2], Jieying Zhang[1], Yukun Qi[2], Lingkai Meng[1], Na Kong[1], Yingjie Chai[3], Qian Hu[1], Zhenyu Xing[1], Wuwei Ren [2] ✉, Fuyou Li [3,4] ✉ & Xingjun Zhu [1] ✉

Nanothermometers enable the detection of temperature changes at the microscopic scale, which is crucial for elucidating biological mechanisms and guiding treatment strategies. However, temperature monitoring of micron-scale structures in vivo using luminescent nanothermometers remains challenging, primarily due to the severe scattering effect of biological tissue that compromises the imaging resolution. Herein, a lanthanide luminescence nanothermometer with a working wavelength beyond 1500 nm is developed to achieve high-resolution temperature imaging in vivo. The energy transfer between lanthanide ions ($Er^{3+}$ and $Yb^{3+}$) and $H_2O$ molecules, called the environment quenching assisted downshifting process, is utilized to establish temperature-sensitive emissions at 1550 and 980 nm. Using an optimized thin active shell doped with $Yb^{3+}$ ions, the nanothermometer's thermal sensitivity and the 1550 nm emission intensity are enhanced by modulating the environment quenching assisted downshifting process. Consequently, minimally invasive temperature imaging of the cerebrovascular system in mice with an imaging resolution of nearly 200 μm is achieved using the nanothermometer. This work points to a method for high-resolution temperature imaging of micron-level structures in vivo, potentially giving insights into research in temperature sensing, disease diagnosis, and treatment development.

Temperature monitoring based on nanothermometers (NTM) has attracted significant attention in recent years[1–3]. Due to the miniaturized dimensions of NTM compared to conventional temperature-detecting devices, NTM can detect microscopic temperatures inside tissues and cells. This capability aids in deciphering physiological processes or disease states, thereby paving the way for the development of potential therapeutic strategies[4–9]. Luminescent nanomaterials are promising candidates for establishing NTM, thanks to their

[1]School of Physical Science and Technology & State Key Laboratory of Advanced Medical Materials and Devices, ShanghaiTech University, 393 Middle Huaxia Road, Shanghai, P.R. China. [2]School of Information Science and Technology, ShanghaiTech University, 393 Middle Huaxia Road, Shanghai, P.R. China. [3]Department of Chemistry & State Key Laboratory of Molecular Engineering of Polymers & Collaborative Innovation Center of Chemistry for Energy Materials, Fudan University, 2005 Songhu Road, Shanghai, P.R. China. [4]Institute of Translational Medicine, Shanghai Jiao Tong University, 800 Dongchuan Road, Shanghai, P.R. China. [5]These authors contributed equally: Yukai Wu, Fang Li. ✉e-mail: renww@shanghaitech.edu.cn; fyli@fudan.edu.cn; zhuxj1@shanghaitech.edu.cn

high sensitivity, rapid signal acquisition, and minimally invasive nature of luminescence signal detection[10–13]. To date, various luminescent nanomaterials have been developed as NTM[3,14,15]. For instance, nanodiamonds have enabled real-time temperature monitoring within *Caenorhabditis elegans*[8]. NTM based on lanthanide luminescence nanoparticles facilitated subcellular organelles thermometry and the guidance of photothermal therapy[16–19]. In addition, triplet-triplet annihilation molecules have been reported to detect tissue temperature during inflammation in mouse models[20]. Recently, near-infrared-emitting quantum dots, including $Ag_2S$ and PbS, have been developed as NTM for in vivo applications, such as identifying the inflammation lesions and monitoring the temperature increase during the thermal ablation of tumor[21–23] (Supplementary Table 1). These advancements underscore the strengths of optical NTM in minimally invasive procedures and its capability for temperature detection at the microscopic scale.

Nevertheless, the temperature monitoring of micron-scale biological structures within large-size organisms using optical NTM is still challenging. The primary obstacle is the scattering effect of biological tissues, which diffuses the optical signal, resulting in poor spatial resolution and hindering the accurate observation of temperature distribution. From a biomedical perspective, the ability to acquire the temperature information of micron-level tissue is both meaningful and necessary, particularly when achieved through minimally invasive methods. For example, brain temperature is a crucial marker for neurological functions, behaviors, and the progression of brain diseases[24–28], which is influenced by local heat production and the temperature of the cerebral vasculature and blood flow[29]. The cerebrovascular system runs through the entire brain, which plays a pivotal role in maintaining brain temperature homeostasis. Instances like deep hypothermia caused by systematic inflammation (e.g., sepsis) can drastically reduce brain temperature via blood flow in the cerebral vessels, impairing brain autoregulation to induce memory loss, altered and loss of consciousness[30–32]. It has been observed that sepsis patients experiencing hypothermia exhibit higher mortality rates than those with hyperthermia[33]. The current temperature measurement techniques include imaging-based thermometry, contact probes, and luminescence nanothermometers[34–38] (Supplementary Table 2). Given the micron-sized scale and delicate nature of the cerebral vessels, it is difficult and risky to measure the cerebral vessel temperature by invasive thermal probe. Moreover, for the application in brain disease diagnosis, the invasive implantation of temperature detection devices is time-consuming, which may delay the treatment and cause unnecessary tissue damage. For imaging-based thermometry, infrared thermal imaging is most commonly used, but it can only assess surface temperature. Magnetic resonance spectroscopy (MRS) and photoacoustic imaging have been utilized to detect the temperature inside the body[34,35]. However, these methods still fall short in accurately mapping the temperature of micron-scale structures, such as small cerebral vessels, due to limited spatial resolution[24]. Therefore, the minimally invasive measurement of temperature within micron-scale cerebral vessels remains a challenge, which impedes the development of brain disease theranostics and neuroscience research.

Near-infrared (NIR) luminescence, as the signal source for optical NTM, is especially advantageous for in vivo applications due to its large tissue penetration depth. Moreover, the emissions within the second and the third NIR biological windows (NIR-II, >1000 nm and NIR-III, >1350 nm) exhibit ideal spatial resolution, so NIR-II/III emissive NTM are expected to conquer the challenge in temperature detection of micron-scale biological structures[39–43]. However, NTM with NIR-III thermometric working wavelengths remains scarce. As a result, it is urgent to develop long-wavelength emissive NTM to achieve the temperature detection of delicate biological structures. Lanthanide-doped nanoparticles known for their excellent photostability, narrow emission bands, and highly tunable wavelengths extending from the

visible to the near-infrared regions up to 1600 nm, are ideal candidates as NIR-II/III NTMs[44–48]. Lanthanide ions, including $Er^{3+}$, $Ho^{3+}$, $Nd^{3+}$, and $Yb^{3+}$ with NIR-II/III emissions, have been employed as emitters in optical NTM for in vivo applications[49–52]. Jaque et al. introduced NIR-II NTM using $Nd^{3+}$ and $Yb^{3+}$ as emitters for temperature sensing[53]. Chen et al. reported $Yb^{3+}$, $Ho^{3+}$, and $Er^{3+}$ codoped nanoparticles and investigated the energy transfer process of those lanthanide ions during temperature sensing to provide a concept of designing lanthanide NTM[49]. Despite these advancements, in vivo temperature monitoring of micron-scale structures using NIR-II/III NTM is still limited, possibly due to the non-ideal working wavelengths or insufficient luminescence intensity for in vivo use.

Herein, we have developed a NIR-III emissive lanthanide ions doped nanocomposite as bidirectional-responsive ratiometric nanothermometer (abbreviated as LIBRA), $NaErF_4$:Yb@$NaYF_4$:Yb. Under the excitation of an 808 nm laser, this nanocomposite can generate 1550 nm emission through $^4I_{13/2} \rightarrow {}^4I_{15/2}$ transition of $Er^{3+}$ and 980 nm emission via the $^2F_{5/2} \rightarrow {}^2F_{7/2}$ transition of $Yb^{3+}$ and $^4I_{11/2} \rightarrow {}^4I_{15/2}$ transition of $Er^{3+}$. $Er^{3+}$ ions serve dual roles as emitters and sensitizers, facilitating photon energy transfer to $Yb^{3+}$ ions. Through optimization of a thin active shell layer doped with $Yb^{3+}$, LIBRA exhibits improved thermal responsive sensitivity and enhanced 1550 nm emission intensity by modulating the interaction between $H_2O$ and lanthanide ions. Both emissions are temperature sensitive and display inverse thermal responses, of which the 1550 nm emission increases and the 980 nm emission decreases with temperature elevation, thereby offering higher thermal sensitivity through the ratio of these two peaks. The mechanism of temperature response of the nanocomposite in aqueous solution is investigated in detail, which can be attributed to energy transfer between lanthanide ions ($Er^{3+}$ and $Yb^{3+}$) and $H_2O$ molecules, named as the environment quenching assisted downshifting (EQAD) process. By utilizing the temperature response behavior and the long-wavelength emission of the nanocomposite, high spatial resolution temperature monitoring of micron-scale cerebral vessels in a hypothermia mouse model was achieved via NIR-III imaging. The correlation between cerebrovascular and brain temperatures during hypothermia progression was also investigated, which will provide valuable insights for future neuroscience studies.

## Results
### Characterization of lanthanide ions doped nanocomposite as bidirectional-responsive ratiometric nanothermometer (LIBRA)
$NaErF_4$:Yb@$NaYF_4$:Yb (LIBRA) core-shell nanocomposite was synthesized via a solvothermal method. $NaErF_4$:20%Yb (Core-LnNP) was designed as the core with high doping concentration of $Er^{3+}$ ions to enhance the 1550 nm emission ($^4I_{13/2} \rightarrow {}^4I_{15/2}$) via concentration-dependent cross-relaxation, simultaneously increasing the absorption cross-section for 808 nm excitation[54]. $Er^{3+}$ ions also played a crucial role in sensitizing the 980 nm emission of $Yb^{3+}$ ($^2F_{5/2} \rightarrow {}^2F_{7/2}$), allowing for dual emissions under 808 nm irradiation with a reduced heating effect. $Yb^{3+}$ ions in the core acted as energy trapping centers for the enhancement of 1550 nm emission[54]. The $NaYF_4$:Yb shell layer repaired the surface defects of Core-LnNP and importantly, prevented the direct non-radiative relaxation between $Er^{3+}$ ions and environmental quenching centers (Fig. 1). Notably, the $Yb^{3+}$-doped shell (active shell), compared to an undoped shell (inert shell), facilitated energy migration to the core nanoparticles, enhancing the thermal sensitivity of ratiometric nanothermometer. Transmission electron microscopy (TEM) revealed that Core-LnNP had a spherical morphology with an average diameter of 22.29 nm (Fig. 2a). The subsequent coating with the $NaYF_4$:Yb shell layer increased the diameter to 25.34 nm and the spherical shape was preserved (Fig. 2b, c), indicating that the shell layer had been combined onto the core nanoparticle through epitaxial growth. High-resolution TEM (HR-TEM) images of Core-LnNP and LIBRA samples confirmed the (1 0 0) lattice planes of hexagonal $NaErF_4$

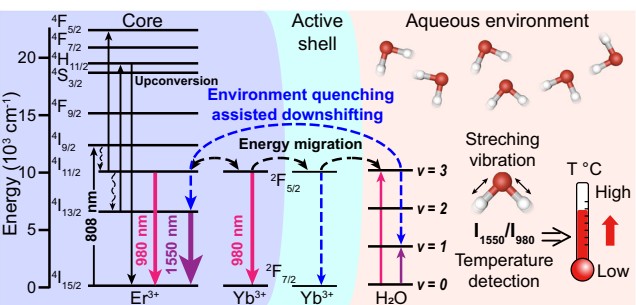

**Fig. 1 | Schematic diagram of the temperature detection mechanism of the nanocomposite.** Lanthanide ions doped nanocomposite as bidirectional ratiometric nanothermometer (NaErF₄:Yb@NaYF₄:Yb, abbreviated as LIBRA) with core-shell structure is developed for temperature imaging in vivo. Under 808 nm excitation, LIBRA generates 1550 nm emission caused by $^4I_{13/2} \rightarrow {}^4I_{15/2}$ transition of $Er^{3+}$ ions and 980 nm emission derived from $^2F_{5/2} \rightarrow {}^2F_{7/2}$ transition of $Yb^{3+}$ ions and $^4I_{11/2} \rightarrow {}^4I_{15/2}$ transition of $Er^{3+}$ ions. Upon temperature elevation, the quenching effects of $H_2O$ stretching vibration transitions on the $^4I_{11/2}$ state of $Er^{3+}$ and $^2F_{5/2}$ state of $Yb^{3+}$ are promoted that induce the decrement of 980 nm emission and increment of 1550 nm emission of LIBRA. This process is utilized for temperature detection in aqueous environment. With the doping of $Yb^{3+}$ in the active shell layer, energy migration among lanthanide ions is achieved to improve 1550 nm emission intensity and enhance thermal sensitivity.

with an approximate 0.52 nm interplanar spacing, showcasing the high crystallinities of both nanoparticles (Fig. 2d, e). X-ray powder diffraction (XRD) analysis affirmed that both nanoparticles were in hexagonal phases, matching well with the standard diffraction patterns of β-NaErF₄ (JCPDS 27-0689) and β-NaYF₄ (JCPDS 28-1192), respectively (Supplementary Fig. 1). Scanning TEM (STEM), energy dispersive X-ray spectroscopy (EDS) element mapping and line scanning analysis (Fig. 2c, f, g) demonstrated a high $Er^{3+}$ doping in the core, $Yb^{3+}$ distribution across core and shell, and minor $Y^{3+}$ presence in the shell, confirming LIBRA's the core-thin-active-shell configuration. For biological applications, the surface of LIBRA was further modified with 1,2-distearoyl-sn-glycero-3-phosphoethanolamine-poly(ethylene glycol) (DSPE-PEG) (denoted as LIBRA@PEG). Fourier transform infrared (FTIR) spectroscopy confirmed the successful assembly of DSPE-PEG on the surface of LIBRA since the C−O bond stretching vibration at 1103 cm⁻¹ in DSPE-PEG could be identified. LIBRA before DSPE-PEG modification exhibited strong absorptions at 2920, 2954 (asymmetric and symmetric stretching vibration of C−H in alkyl chain)[55], 1558, and 1457 cm⁻¹ (asymmetric and symmetric stretching vibrations of carboxylic group), which came from oleic acid (OA) ligands (Fig. 2h). Dynamic light scattering (DLS) and TEM assessments of LIBRA@PEG in various media (deionized water, phosphate-buffered saline (PBS) and fetal bovine serum (FBS)) showed good dispersity without obvious aggregation, ideal for biomedical uses (Supplementary Fig. 2).

The high $Er^{3+}$ ions doping promoted excitation energy utilization for NIR luminescence and enhanced the energy transfer to $Yb^{3+}$ ions, although luminescence quenching by solvents or ligands remained a concern with heavy doping designs[56]. Luminescence spectroscopy confirmed that the NaYF₄:Yb shell layer on LIBRA effectively countered quenching, with NIR luminescence spectra indicating a substantial enhancement in NIR emissions of LIBRA when dispersed in aqueous solution and modified with DSPE-PEG for hydrophilization. Specifically, LIBRA with NaYF₄:50%Yb shell exhibited a 23-fold increase in 1550 nm emission intensity compared to Core-LnNP (Fig. 2i), highlighting the anti-quenching effect of shell layer.

## Temperature response behavior and mechanism of LIBRA

To investigate the temperature response behavior of the nanocomposite, LIBRA@PEG was dispersed in deionized water and subjected to a series of temperatures from 283 to 363 K, controlled by an

external heating device. The NIR luminescence spectra of LIBRA@PEG were collected simultaneously under the excitation of an 808 nm laser. Specifically, in LIBRA@PEG sample with 50% of $Yb^{3+}$ doped in the shell, the 1550 nm emission intensity increased consistently while the 980 nm emission intensity decreased with rising temperature (Fig. 3a, b and Supplementary Fig. 3). The ratio of luminescence intensities at 1550 and 980 nm (ratio ($I_{1550}/I_{980}$)) versus temperature exhibited a linear behavior from 283 to 363 K (Fig. 3c), which can be described as $I_{1550}/I_{980} = 96.7 \times 10^{-3}\,T - 22.8$ ($T$ given in K). The intensity changes of the two emission bands responsive to temperature were in the reverse direction, which was called "bidirectional". This led to a higher rate of change in the ratio than in the intensity of a single emission, yielding a higher thermal sensitivity (Fig. 3a, d). The relative thermal sensitivities ($S_r$) of the ratio ($I_{1550}/I_{980}$) and the individual emissions ($I_{1550}$ and $I_{980}$, respectively) at varying temperatures were illustrated in Fig. 3d. The $S_r$ of the ratio was higher than that of either single emission band ($I_{1550}$ or $I_{980}$) at each temperature point. The typical $S_r$ of ratio at 303 K (30 °C) was 1.51% K⁻¹. LIBRA with various doping concentrations of $Yb^{3+}$ ions in the shell layer, ranging from 0 to 100% mol, were prepared to examine their impact on thermal sensitivity (Supplementary Fig. 4). LIBRA with 50% $Yb^{3+}$ doping in the shell achieved the highest $S_r$ and maintained 80% of the NIR-III emission intensity compared to samples without $Yb^{3+}$ doping in the shell (Supplementary Figs. 5 and 6). As shown in Fig. 3e, LIBRA exhibited high repeatability at 99.3% across four heating-and-cooling cycles. The repeatability is calculated according to Eq. 1,

$$\text{Repeatability} = 1 - \frac{\max(|LIR_c - LIR_i|)}{LIR_c} \quad (1)$$

where $LIR_c$ is the mean value of the luminescence intensity ratio (LIR) and $LIR_i$ is the measured value each time[57]. Moreover, the change of excitation power density had no apparent influence on LIR (Fig. 3f). The temperature uncertainty ($\Delta T$) of LIBRA@PEG is defined according to Eq. 2,

$$\Delta T_{\min} = \frac{\sigma}{S_a} \quad (2)$$

where $\sigma$ is the standard deviation of luminescence intensity ratio and $S_a$ is the absolute sensitivity that equals to the slope of the fitting curve in the plot of luminescence intensity ratio versus temperature (Fig. 3c). The $\Delta T$ values of LIBRA@PEG remained low, which were calculated to be 0.05 to 0.105 K at the range of 283 to 363 K (Supplementary Fig. 7). In addition, LIR kept stable across different concentrations of LIBRA in aqueous dispersion, indicating that the concentration and potential aggregates do not affect the thermal readout (Supplementary Fig. 8). Based on the results above, it can be concluded that LIBRA with the bidirectional response, high repeatability and excitation power independence would effectively improve the temperature detecting performance, which was helpful in biological applications.

Given the interesting temperature response of LIBRA, further investigation into the underlying mechanism was carried out (Fig. 4a). Firstly, to investigate the intrinsic temperature response of LIBRA, the emission spectra of LIBRA in the solid powder state without any solvent were measured under a nitrogen atmosphere. As shown in Fig. 4b, c, the 980 nm emission exhibited a mild decrease (14%) in intensity from 10 to 90 °C, while the 1550 nm emission slightly increased as temperature elevated. The decrement of 980 nm emission could be attributed to the thermal quenching of the nanocrystals since the thermally populated phonon relaxation could promote the non-radiative transition of excited energy level[58]. The minor increment of 1550 nm emission may be caused by the multi-phonon relaxation (MPR) from $^4I_{11/2}$ to $^4I_{13/2}$ of $Er^{3+}$ ions[49,58]. Dimethylfuran (DMF) with good thermal stability is well-suited as a solvent for dispersing lanthanide-doped nanoparticles. We then explored the thermal

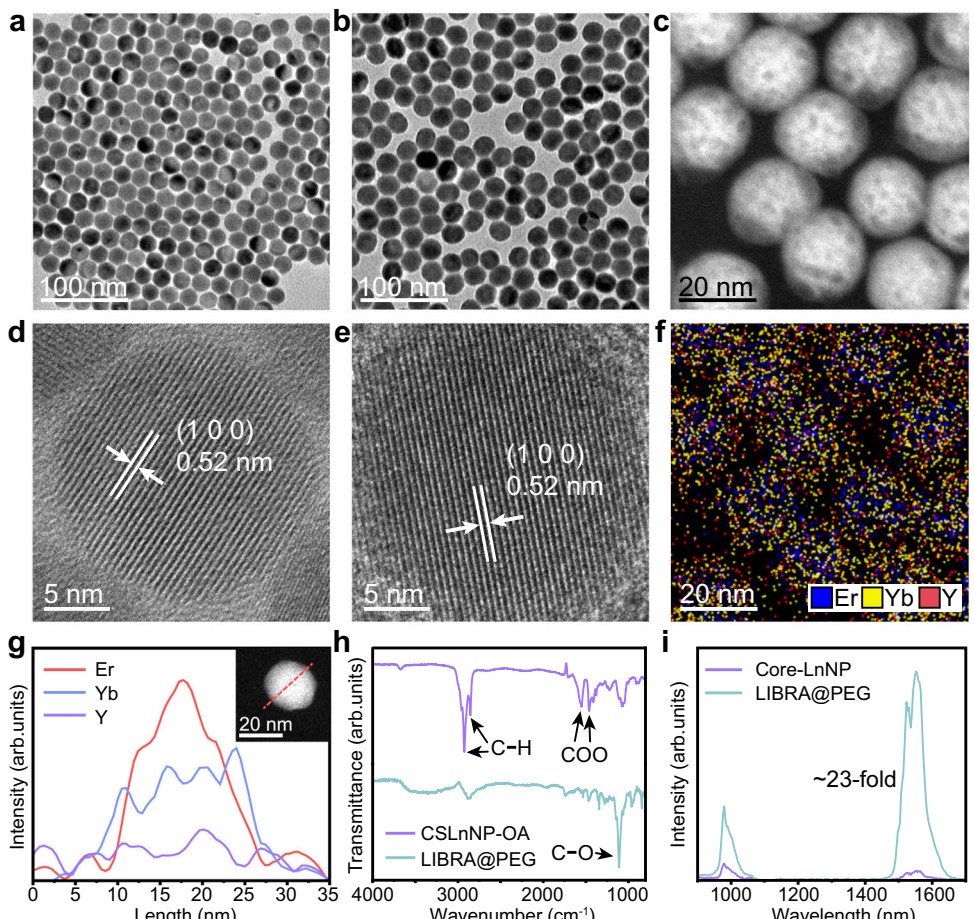

**Fig. 2 | Fundamental characterizations of LIBRA.** Transmission electron microscopy (TEM) images of **a** NaErF$_4$:Yb (Core-LnNP), **b** NaErF$_4$:Yb@NaYF$_4$:Yb (LIBRA). TEM images in (**a**) and (**b**) represent the similar results of synthesis experiments repeated independently for three times. **c** Scanning transmission electron microscopy (STEM) image of LIBRA. STEM image represents the similar results of synthesis experiments repeated independently for three times. High-resolution TEM (HR-TEM) images of **d** Core-LnNP and **e** LIBRA nanoparticle. The lattice planes of (1 0 0) with 0.52 nm interplanar spacing of hexagonal NaErF$_4$ is identified in (**d**) and (**e**). HR-TEM images in (**d**) and (**e**) represent the similar results of synthesis experiments repeated independently for three times. **f** Energy dispersive X-ray spectroscopy (EDS) elemental mapping of LIBRA, showing the distributions of Er (blue dots), Yb (yellow dots), and Y (red dots) elements in the core and shell layers of the nanocomposite. EDS elemental mapping image represents the similar results of synthesis experiments repeated independently for three times. **g** Representative EDS line scan analysis of LIBRA. Inset, STEM image of the nanoparticle performed with EDS line scan. EDS line scan and STEM results represent the similar outcomes of synthesis experiments repeated independently for three times. **h** Fourier transform infrared (FTIR) spectra of the as-synthesized LIBRA capped with oleic acid (CSLnNP-OA) and after 1,2-distearoyl-sn-glycero-3-phosphoethanolamine-poly(ethylene glycol) (DSPE-PEG) modification (LIBRA@-PEG). **i** Near-infrared (NIR) luminescence spectra of Core-LnNP and LIBRA modified with DSPE-PEG in aqueous solutions. The protection effect of NaYF$_4$:Yb shell layer results in a nearly 23-fold enhancement of NIR emissions in LIBRA@PEG compared to Core-LnNP.

response behavior of LIBRA in DMF. The luminescence decay curve indicated that DMF only showed weak quenching effects to the emissions at 980 and 1550 nm (Fig. 4d) and the thermal response behavior of 980 nm emission of LIBRA dispersed in DMF was similar to the case in solid state (Supplementary Fig. 9). Meanwhile, the 1550 nm emission was nearly unchanged with temperature (Supplementary Fig. 9). DMF has relatively weak absorption near 1500 nm (Supplementary Fig. 10), of which the peak edge has a moderate overlap with the 1550 nm emission of LIBRA. In this case, the minor increment of 1550 nm emission observed in the solid powder state was further reduced in DMF. The thermal sensitivity in aqueous environment was much more pronounced than that in solid powder state (Fig. 4c). If the dispersion was switched to water, the intensity of 980 nm emission dropped more drastically to 64% and 1550 nm emission went up quickly as temperature rising (Fig. 4b, c).

This phenomenon evidenced the pivotal role of H$_2$O molecules in the temperature response of LIBRA. To decouple the contributions of Yb$^{3+}$ and Er$^{3+}$ ions in 980 nm emission and study the interaction between Er$^{3+}$ ions and H$_2$O, NaErF$_4$:20%Y@NaYF$_4$ nanoparticles were synthesized with identical morphology and size as LIBRA (Supplementary Fig. 11) and their emission spectra were measured in aqueous environment under varied temperatures (Supplementary Fig. 12). As the temperature rose, the 980 nm emission corresponding to $^4I_{11/2} \rightarrow {}^4I_{15/2}$ transition of Er$^{3+}$ decreased while the 1550 nm emission corresponding to $^4I_{13/2} \rightarrow {}^4I_{15/2}$ transition increased. This demonstrated a similar phenomenon to that observed in LIBRA (NaErF$_4$:Yb$^{3+}$@NaYF$_4$:Yb$^{3+}$). It is known that H$_2$O molecules have O−H stretching vibration modes (ν), such as the vibration transition ν = 0 → ν = 1 (3400–3500 cm$^{-1}$), of which the energy gap matches well with that of the $^4I_{11/2} \rightarrow {}^4I_{13/2}$ (~3700 cm$^{-1}$) transition of Er$^{3+}$ ion[59]. In addition, based on the result of Raman spectroscopy, the absorption of stretching vibration of H$_2$O rises with temperature[60]. Therefore, it can be inferred that the energy transfer would happen between H$_2$O and Er$^{3+}$ ions. With the elevation of temperature, the energy transfer was facilitated due to increased resonance match so the emission intensities of LIBRA changed accordingly[59]. To further confirm the proposed mechanism, luminescence lifetime of NaErF$_4$:20% Y@NaYF$_4$ at 980 and 1550 nm were measured at different temperatures in aqueous solution (Supplementary Fig. 13). It showed that the lifetime

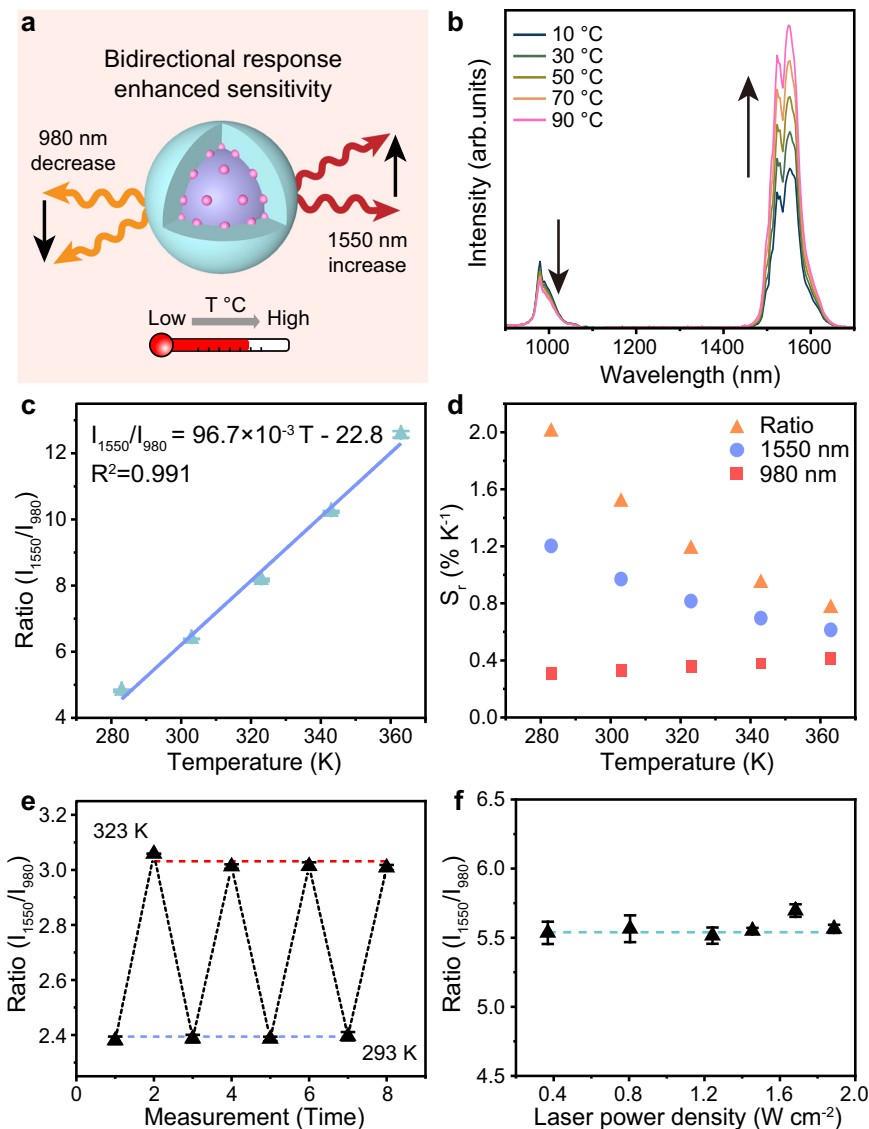

**Fig. 3 | Temperature response behavior of LIBRA. a** Schematic of temperature response of LIBRA. The sensitivity of temperature response can be enhanced by bidirectional changes of the luminescence intensities at 980 and 1550 nm. **b** NIR luminescence spectra of LIBRA@PEG in aqueous dispersion at 10 to 90 °C by external heating. The emission at 980 nm decreased and the emission at 1550 nm increased with temperature elevation. **c** The ratio of emissions at 1550 and 980 nm (*Ratio* ($I_{1550}/I_{980}$)) *versus* temperature (given in K) for temperature calibration. Data were presented as mean values based on three individual measurements (n = 3) by spectrometry. Error bars were defined as standard deviation. **d** Relative thermal sensitivity ($S_r$) of LIBRA@PEG based on ratio of 980 and 1550 nm emissions and single emission intensity. The temperature range was from 283 to 363 K. **e** Repeatability of the ratio of emissions at 1550 and 980 nm (*Ratio* ($I_{1550}/I_{980}$)) with 4 cycles of heating and cooling between 293 and 323 K. Data were presented as mean values based on three individual measurements (*n* = 3) by spectrometry. Error bars were defined as standard deviation. **f** Ratio of emissions at 1550 and 980 nm (*Ratio* ($I_{1550}/I_{980}$)) under the excitation of 808 nm laser with varied power densities at room temperature. Data were presented as mean values based on three individual measurements (*n* = 3) by spectrometry. Error bars were defined as standard deviation.

of 980 nm emission decreased obviously upon heating, implying that $Er^{3+}$ at $^4I_{11/2}$ state may undergo relaxation processes and their populations were reduced. On the other hand, the lifetime of 1550 nm emission increased very slowly, which may be caused by the increased populations of the $^4I_{13/2}$ state with temperature elevation. Considering the changes of luminescence intensity and lifetime, we have reasons to believe that the rise of temperature may promote the energy transfer between $Er^{3+}$ at $^4I_{11/2}$ state and $H_2O$ at ground state to induce $^4I_{11/2} \rightarrow {}^4I_{13/2}$ transition of $Er^{3+}$ (denoted as Coupling 1: $^4I_{11/2}$ ($Er^{3+}$) + v = 0 ($H_2O$) $\rightarrow {}^4I_{13/2}$ ($Er^{3+}$) + v = 1 ($H_2O$)). The increased population of $^4I_{13/2}$ state can explain the enhancement of 1550 nm emission and the decreased population of $^4I_{11/2}$ state contributed partially to the suppression of 980 nm emission at high temperature (Supplementary Fig. 14). The similar change in luminescence lifetime with

temperature increase was also observed (Supplementary Fig. 15). Based the results above, the thermal response mechanism of LIBRA could be attributed to the temperature-sensitive interaction between lanthanide ions and water molecule, which was termed as environment quenching assisted downshifting (EQAD).

Moreover, water molecules also have O−H overtone transition (v = 0 → v = 3) at ~10,300 $cm^{-1}$, of which the population increases with temperature rising (Supplementary Fig. 16). The energy gap of this transition matches well with that of the $^2F_{5/2} \rightarrow {}^2F_{7/2}$ (~10,200 $cm^{-1}$) transition in $Yb^{3+}$ ions (Fig. 1)[61-63]. Hence, a similar temperature-responsive energy transfer process would also happen between $Yb^{3+}$ ion and $H_2O$ (denoted as Coupling 2: $^2F_{5/2}$ ($Yb^{3+}$) + v = 0 ($H_2O$) $\rightarrow {}^2F_{7/2}$ ($Yb^{3+}$) + v = 3 ($H_2O$))[63], which was another possible reason for the decrease of 980 nm emission. Nanocomposite samples with inert shell

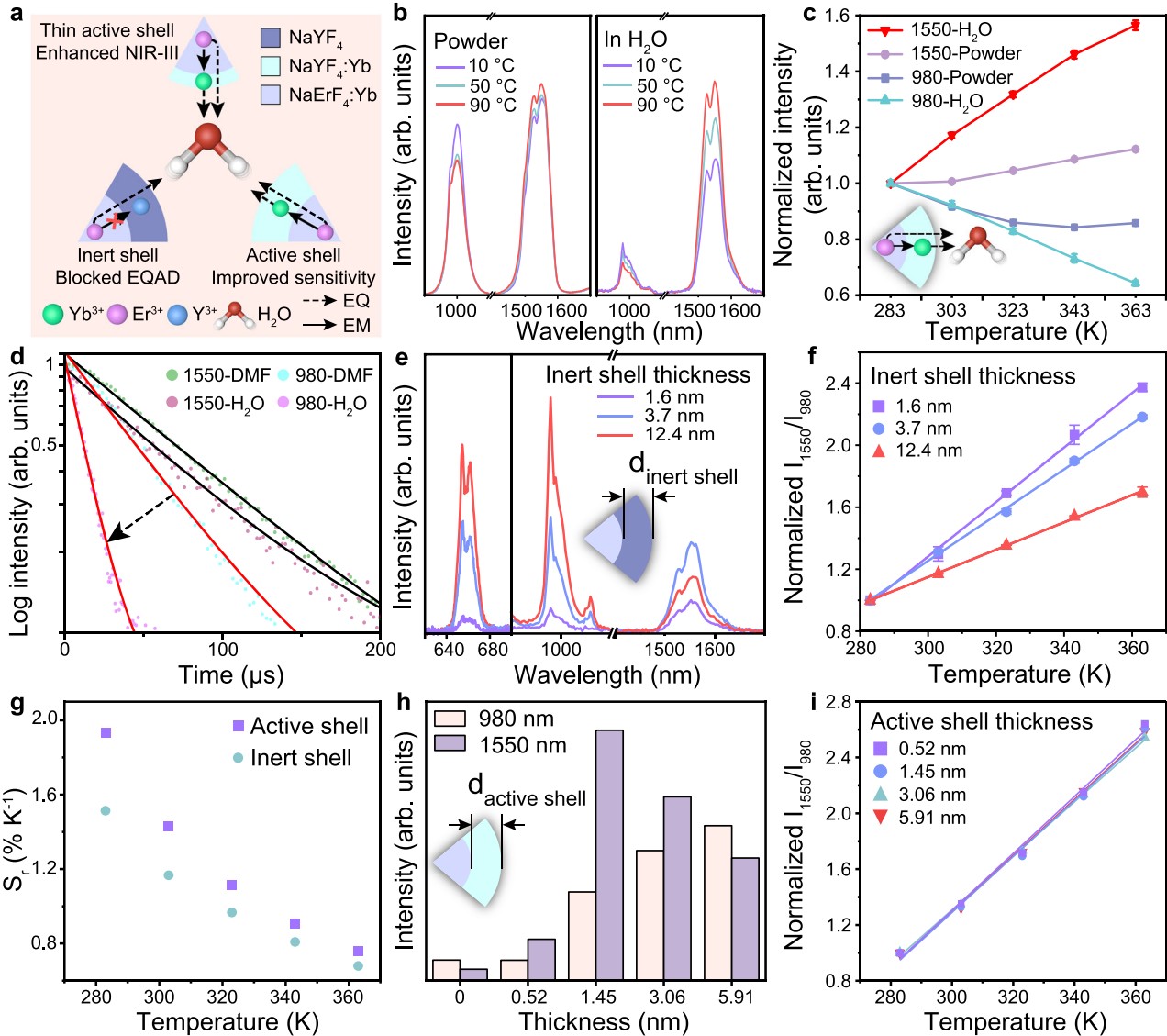

**Fig. 4 | Investigation of temperature response mechanism of LIBRA.**
**a** Schematic of LIBRA with enhanced sensitivity and the third near-infrared biological window (NIR-III) emission endowed by the optimized thin active shell doped with $Yb^{3+}$ ions to achieve the energy migration (EM) of the lanthanide ion excited states across the core and shell layer of the nanocomposite and the environment quenching (EQ) by $H_2O$ molecules of the excited states. **b** NIR luminescence spectra of LIBRA in solid powder and those dispersed in water at 10, 50, and 90 °C.
**c** Luminescence intensity change of the emissions at 980 and 1550 nm of LIBRA in solid powder (980-Powder and 1550-Powder) and dispersed in water (980-$H_2O$ and 1550-$H_2O$) at different temperatures. The intensities were normalized at 283 K to compare the changing behavior of different emissions. Data were presented as mean values based on three individual measurements ($n = 3$) by spectrometry. Error bars were defined as standard deviation. **d** Luminescence decay curves of emissions at 980 and 1550 nm of LIBRA@PEG with 50% $Yb^{3+}$ in the shell at different temperatures in water (980-$H_2O$ and 1550-$H_2O$) and in dimethylformamide (980-DMF and 1550-DMF). **e** Visible and NIR luminescence spectra of LIBRA@PEG with the inert shell thickness of 1.6, 3.7, and 12.4 nm in aqueous environment.
**f** Luminescence intensity ratio change of the emissions at 1550 and 980 nm ($I_{1550}/I_{980}$) of LIBRA@PEG with various inert shell thicknesses at different temperatures. The ratios were normalized at 283 K to compare the slopes of fitted curves of different nanocomposites. Data were presented as mean values based on three individual measurements ($n = 3$) by spectrometry. Error bars were defined as standard deviation. **g** Relative thermal sensitivity ($S_r$) of ratio ($I_{1550}/I_{980}$) of LIBRA@PEG with 0 and 50% $Yb^{3+}$ in the shell. **h** Luminescence intensity change of the emissions at 980 and 1550 nm of LIBRA@PEG with the active shell thickness of 0, 0.52, 1.45, 3.06, and 5.91 nm in aqueous environment. **i** Luminescence intensity ratio change of emissions at 1550 and 980 nm ($I_{1550}/I_{980}$) of LIBRA@PEG with different active shell thickness at different temperatures. The ratios were normalized at 283 K to compare the slopes of fitted curves of different nanocomposites.

($Yb^{3+}$ doping in core only and without $Yb^{3+}$ doping in the shell) and active shell ($Yb^{3+}$ doping in both core and shell layer) were used to examine the impact of $Yb^{3+}$ ions on temperature response. The luminescence lifetime at 980 nm for the nanocomposite with an active shell was obviously shorter than that with an inert shell (Supplementary Fig. 17), illustrating that the energy migration among $Yb^{3+}$ ions across the core and shell could enhance Coupling 2, attributed to closer proximity of $Yb^{3+}$ ions to water molecules. This mechanism resulted in higher $S_r$ of 980 nm emission of LIBRA with an active shell than the one

with an inert shell (Supplementary Fig. 18), thereby improving the $S_r$ of luminescence intensity ratio ($I_{1550}/I_{980}$).

In addition, the influence of the distance between lanthanide ions and $H_2O$ molecules on the temperature-responsive energy transfer was investigated. Nanocomposites with the inert shell thickness of 1.6, 3.7, and 12.4 nm were synthesized by epitaxial growth and the corresponding TEM images were shown in Supplementary Fig. 19. It was observed that as the shell thickness increased, both the 660 nm (upconversion luminescence) and 980 nm emissions intensified, while

the 1550 nm emission weakened (Fig. 4e), a phenomenon previously described by Fisher et al. as the surface quenching assisted down-shifting (SQAD) mechanism[64]. The emission ratio ($I_{1550}/I_{980}$) *versus* temperature exhibited a decreased slope with increased shell thickness (Fig. 4f), indicating that a greater distance between lanthanide ions and $H_2O$ molecules hindered the energy transfer processes (Coupling 1 and Coupling 2). In nanocomposites with a thicker inert shell, the populations of $^4I_{11/2}$ ($Er^{3+}$) and $^2F_{5/2}$ ($Yb^{3+}$) states were stable, resulting in lesser changes in the LIR with temperature compared to those with an active shell (Fig. 4g). The study also explored how the thickness of the active shell influences NIR-II/III emission intensity and temperature response behavior. As shown in Supplementary Fig. 20, a series of LIBRA samples with different active shell thicknesses of 0, 0.52, 1.45, 3.06, and 5.91 nm were synthesized. As depicted by Fig. 4h and Supplementary Fig. 21, 980 nm emission progressively increased with the thickness of active shell. For the nanocomposites with active shell thicknesses less than 1.45 nm, the intensity of 1550 nm emission enhanced with increasing shell thickness. However, for shells thicker than 1.45 nm, the 1550 nm emission decreased as shell thickness increased. This effect could be attributed to surface defects causing significant energy dissipation and surface quenching for thinner shells[12]. When shell thickness was greater than 1.45 nm, the Coupling 1 process was suppressed by the increased shell thickness, illustrating a balance between surface quenching and EQAD for 1550 nm emission. Notably, LIBRA with an active shell thickness of 1.45 nm exhibited the strongest 1550 nm emission, indicating an optimal balance. Interestingly, the thickness of the active shell did not markedly affect the relative thermal sensitivity for thermometry, contrasting with the effects observed for different inert shell thicknesses (Fig. 4h, i).

## NIR-III cerebral vessel imaging and temperature detection using LIBRA

The biosafety and potential toxicity of nanoparticles are related to the interaction between biological system and the physicochemical properties of nanoparticles, such as chemical composition, dissolution rate, surface state, and physical agglomeration. For lanthanide nano-materials, it is reported that lanthanide ions are not highly toxic[65,66]. Surface modifications, such as coating with polyethylene glycol (PEG), and the use of matrices with low dissolution rates (like sodium lan-thanide fluoride), can minimize the release of lanthanide ions and reduce particle agglomeration. These modifications also decrease protein adsorption on the nanoparticle surface, mitigating the immune response. To assess the biological safety of LIBRA@PEG, both in vitro and in vivo toxicity evaluations were conducted. The cyto-toxicity was examined using methyl thiazolyl tetrazolium (MTT) assay. The results indicated that LIBRA@PEG did not have obvious toxicity even under a high incubation concentration of 800 μg ml$^{-1}$, with cell viability of ~85% (Supplementary Fig. 22). In addition, hematoxylin and eosin (H&E) staining analysis was performed to investigate the toxicity of LIBRA@PEG in mice. The main organs (heart, liver, spleen, lung and kidney) of mice were collected 7 days after intravenous injection of LIBRA@PEG. As shown in Supplementary Fig. 23, compared to the control group without LIBRA@PEG injection, no noticeable lesions or injuries was observed in mice treated with LIBRA@PEG, suggesting that the administration of LIBRA@PEG into mice would not induce significant toxicity. The biodistribution of LIBRA@PEG was also investigated in a mouse model. The main organs (heart, liver, spleen, lung, and kidney) were examined for NIR-II imaging 24 h after injection with LIBRA@PEG. As shown in Supplementary Fig. 24, NIR-II lumines-cence signals were predominantly observed in liver and spleen, indi-cating that LIBRA@PEG mainly accumulated in these organs. The nanoparticles administered by intravenous injection can be cleared out from blood circulation gradually by the immune system, which suggests that the luminescence signals in blood vessels will change over time. For example, in a previous report, luminescence signals of

gold clusters in mouse cerebral vessel declined sharply to half within 5 s[67]. In our study, it was found that the half-life of LIBRA@PEG in blood circulation of control mouse was 1.47 h (Supplementary Fig. 25), indi-cating that LIBRA@PEG had less change in luminescence intensity than previous probes over short time periods. Considering that the time for switching two filters to collect 980 and 1550 nm emissions in imaging system is about 2 s, it is necessary to ensure that the signal of LIBRA@PEG is stable in the blood circulation system for more than 2 s. Mice without any treatment (control group) and the ones with lipo-polysaccharide (LPS) injection to induce hypothermia model (hypo-thermia group) were used to study the blood circulation stability of LIBRA@PEG. As depicted in Fig. 5a, b, luminescence signals kept relatively stable in cerebral vessels of mice and the signal changes were as low as 2.3% and 4.7% within 120 s for LPS-induced hypothermia group and control group, respectively, providing sufficient data col-lection time by switching the filters. Moreover, as shown in the Fig. 5a and Supplementary Fig. 26, luminescence signals in cerebral vessels of hypothermia group dropped slowly to 89% while the signals in control group dropped quickly to 70% at 60 min. This phenomenon indicated that LPS-induced hypothermia resulted in longer retention of the nanocomposite in the blood circulation, which can be attributed to the declined cerebral blood flow caused by hypothermia[68]. A representa-tive NIR-III and NIR-II cerebral vessel imaging was shown in Fig. 5c and Supplementary Fig. 27, respectively. With the high spatial resolution of optical imaging beyond 1500 nm, the inferior cerebral veins (ICV), the superior sagittal sinus (SSS), superficial veins (SV), and the transverse sinus (TS) could be clearly identified. More importantly, the smallest diameter of vessel was 202 μm according to Gaussian simulation of plot profiles (Fig. 5d). Under this imaging condition (808 nm laser, 120 mW cm$^{-2}$), we investigated the photothermal effect of excitation light (Supplementary Fig. 28). The temperature elevation of scalp was only 0.53 °C for over 60 s exposure, which ensured its low influence on thermometry of cerebral vessels. The NIR-III imaging with superior spatial resolution puts a solid foundation for temperature detection of micron-scale cerebral vessels in minimally invasive and micro-scopic way.

A hypothermia model in mice, induced by intraperitoneal injec-tion of LPS, was employed to evaluate the temperature measurement capabilities of LIBRA@PEG in the living body (Fig. 6a). Initially, a thermocouple was implanted into the mouse brain to monitor brain tissue temperature before and during the onset of LPS-induced hypothermia (Supplementary Fig. 29). Concurrently, the tempera-tures of the scalp and abdomen skin were recorded using a thermal camera and a thermocouple, respectively (Fig. 6b–d and Supplemen-tary Fig. 30). At the beginning, the temperatures of brain tissue, scalp and abdomen skin were stabilized at ~37 ± 1 °C for 1.5 h. LPS was then intraperitoneally injected to mice (n = 3) to induce hypothermia, resulting in a gradual decrease of approximately 8.5 °C in the tem-peratures of brain tissue, scalp, and abdomen skin about 8 h post-injection. This decrease in temperature can be linked to diminished brain metabolism due to reduced cerebral blood flow and glucose uptake[68]. For the detection of cerebral vessel temperatures, LIBRA@-PEG was administered intravenously to the mice. NIR-II/III imaging was then performed using 900 nm and 1400 nm long-pass filters, respec-tively, for the LIR-based temperature detection. Given the impact of biological tissue on luminescence intensity due to light extinction effects (absorption and scattering) and the limitations of other optical thermometric methods such as lifetime detection for cerebrovascular temperature imaging ("Discussion", Supplementary Figs. 31 and 32), a light propagation model based on photon diffusion equation[69] with Green's function as a solution was introduced in this work so that a calibrated equation was provided to accurately describe the correla-tion of LIR and temperature in biological tissue ("Methods", Supple-mentary Notes 1 and Supplementary Fig. 33). The feasibilities of the light propagation model and the calibrated equation were validated by

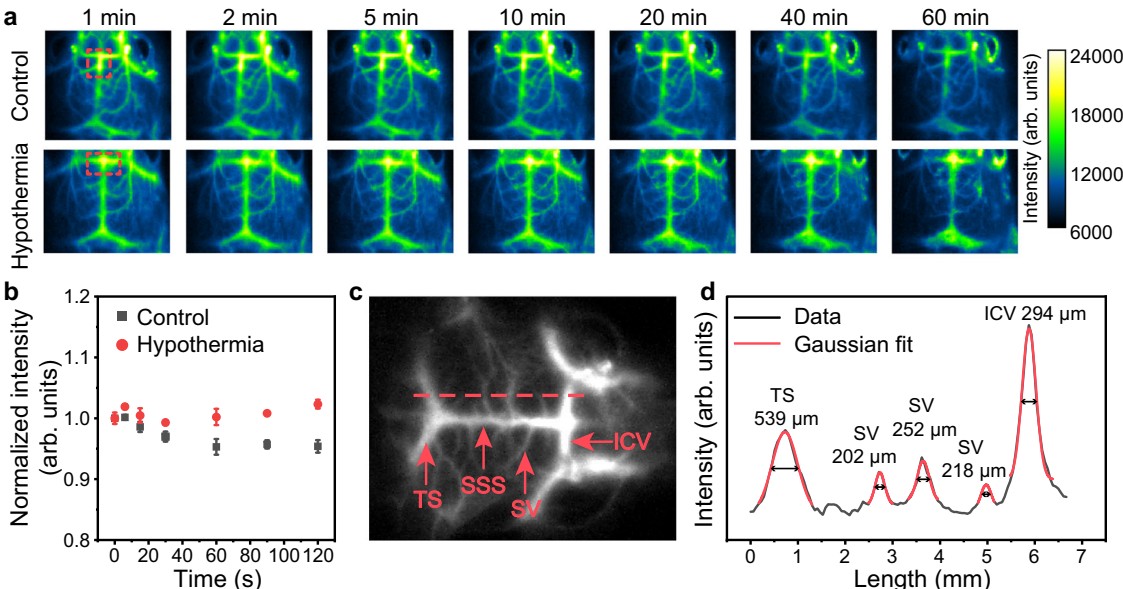

**Fig. 5 | Cerebral vessels imaging with high spatial resolution and stability in the NIR-III region. a** Time-sequence cerebral vessels imaging of control and hypothermia mice under an excitation of 808 nm laser at 120 mW cm$^{-2}$ with 1500 nm long pass filter and an exposure time of 300 ms. **b** Time-dependent quantitative cerebral vessel luminescence intensities (red dash square) of control mouse and hypothermia mouse. The intensities were normalized at 0 s to compare the changing behavior of the two groups. Data were presented as mean values based on three continuous images (*n* = 3). Error bars were defined as standard deviation. **c** Representative cerebral vessels image under the same condition as (**a**). The vessels including inferior cerebral veins (ICV), superior sagittal sinus (SSS), superficial veins (SV), and the transverse sinus (TS) were labeled in red arrows. **d** Plot profile data (black curve) recorded along the red dash lines in (**c**). Gaussian fittings (red curves) of the peaks in the profile provided the full width at half maximum (FWHM) values of ICV, SSS, SV, and TS in NIR-III imaging.

fitting the correlation of LIR and temperature in a series of phantoms (Supplementary Fig. 34) with determined absorption coefficient ($\mu_a$) and reduced scattering coefficient ($\mu'_s$) via NIR imaging (Supplementary Table 3 and Supplementary Figs. 35 and 36). Then, the validated equation was employed to measure the temperature of cerebral vessels in mice based on the imaging results obtained by the emissions at 980 and 1550 nm of LIBRA@PEG and the extinction coefficients of the biological tissues[70,71]. The biological tissues including scalp, skull, etc. covering the veins were of identical thicknesses, which were measured to be 0.69 ± 0.02 mm in this work, so it was inferred that these veins had similar tissue depths and extinction coefficient conditions. Mice before LPS injection (control group) and after LPS injection to induce hypothermia (hypothermia group) exhibited different temperatures in the cerebral blood vessels through NIR imaging using LIR, which confirmed that LIBRA@PEG can discriminate the temperature in the living body for hypothermia diagnosis (Fig. 6e). The average temperatures of ICV, SSS and TS in control group were 37.1, 37.2, and 36.4 °C, respectively, and the ones in hypothermia group were 29.1, 28.7, and 28.3 °C (Fig. 6f), respectively. The obvious temperature decreases of cerebral vessels monitored by NIR-III imaging revealed that the cold blood flow is an important factor for dissipating the heat in the brain region during the LPS-induced sepsis, which gives rise to hypothermia (Fig. 6d).

## Discussion

In conclusion, we have developed a lanthanide ions doped nanocomposite as bidirectional-responsive ratiometric nanothermometer (NaErF$_4$:Yb@NaYF$_4$:Yb, abbreviated as LIRBA) for the high-resolution temperature imaging of micron-scale cerebrovascular system in mice. The temperature-sensitive behavior of LIBRA is attributed to the interaction between lanthanide ions and water molecules, a process facilitated by environment quenching assisted downshifting (EQAD). Enhanced by an optimized thin active shell, LIBRA exhibits improved relative thermal sensitivity and NIR-III luminescence intensity, making it highly effective for temperature mapping in cerebral vessels within a

hypothermia model. The main contribution and innovation of this work lies in introducing a visualized temperature sensing technique that leverages near-infrared emissive nanomaterials for detecting temperature changes in microscopic structures in vivo, such as cerebral blood vessels. To achieve this goal, the use of LIR is necessary and to some extent indispensable because LIR has the merits of rapid detection speed, and steady-state excitation and signal acquisition with sufficient signal intensity for high-resolution imaging. In the case of the temperature imaging of cerebral blood vessels, it is important to conduct the thermometry within a time frame as short as possible because the clearance of the luminescent imaging agent in the circulatory system due to metabolism process will result in the decrease of luminescence intensity as time goes, leading to the deviation of temperature readout. LIBRA presented in this work showed minimal variation of optical signals in vivo over 120 s (Fig. 5b) and was designed with the ratiometric strategy (quick temperature detection), making it an ideal candidate for detecting the temperature of cerebral vessels. Moreover, to address the challenge of light extinction by biological tissues affecting LIR in vivo, the light propagation model used in optical tomography was employed to calibrate luminescence intensity for improving the accuracy of temperature detection in vivo.

Lifetime decoding, which relies on the periodic transient-state excitation and capturing time-dependent decay of luminescence intensity signals, requires a prolonged period to gather enough signals for accurate lifetime measurement. Given that the detecting signal of lifetime decoding also depends on the luminescence intensity, lifetime-based nanothermometry due to its relatively slow detection speed is less suitable for temperature detection of cerebral vessels. This is because the metabolism of luminescent probe in the bloodstream will reduce the luminescence intensity, which will affect the accuracy of lifetime decoding and thus bring about additional deviations (Supplementary Fig. 31). Furthermore, the lifetime-based reading derived from luminescence intensity can also be influenced by the absorption and scattering of bio-tissue at a certain extent. For instance, luminescence lifetime imaging using the 1550 nm emission

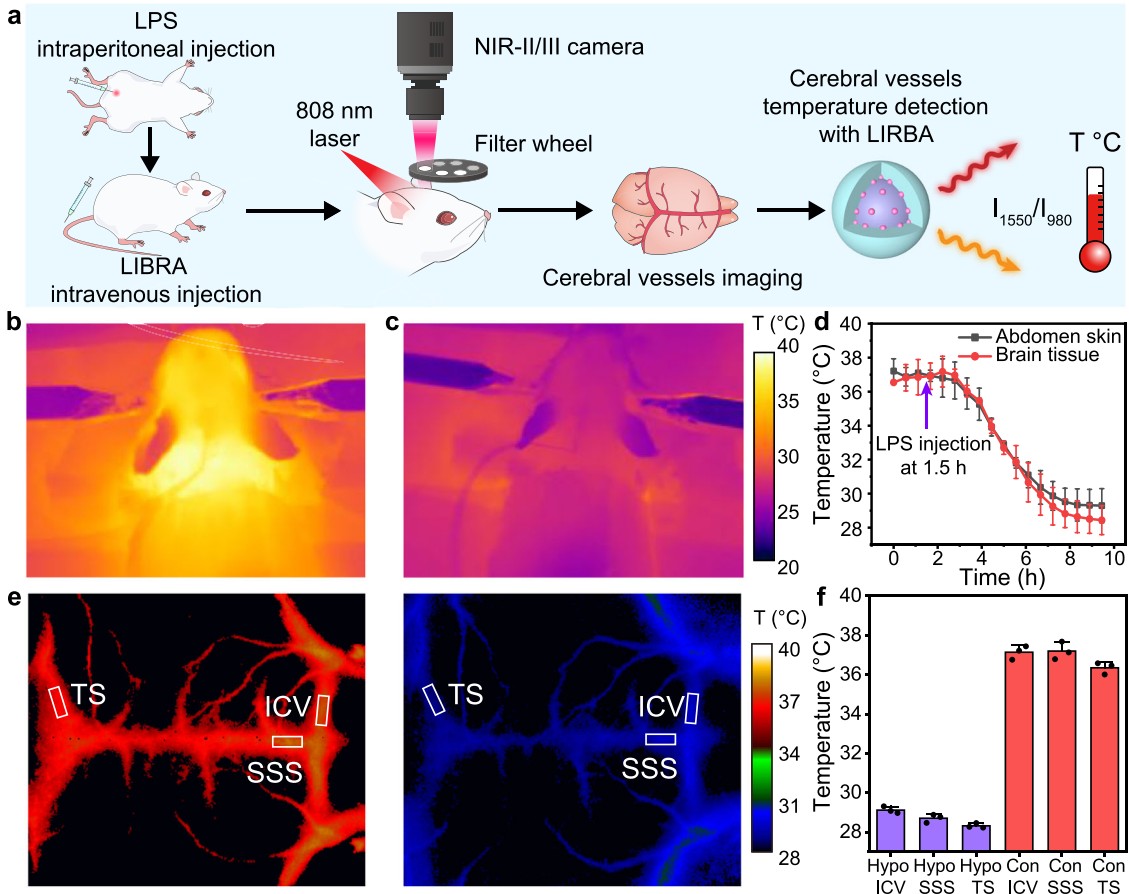

**Fig. 6 | Cerebral vessels temperature detection using LIBRA in hypothermia and control mice. a** Schematic of cerebral vessel temperature detection using LIBRA. Mice were treated with intraperitoneal injection of lipopolysaccharide (LPS) to induce hypothermia and then were injected with LIBRA intravenously. Bioimaging at the second and third near-infrared biological windows (NIR-II/III) was performed in LIBRA-injected mice with camera under 808 nm laser excitation. The luminescence intensity ratio of the emissions at 1550 and 980 nm obtained from bioimaging was used to achieve the ratiometric temperature imaging of the mice without LPS injection as control and hypothermia mice after LPS injection. Thermal images of mice' scalp detected by infrared thermal camera in **b** control and **c** hypothermia mice. **d** Temperature variations of brain tissue and abdomen skin recorded by thermocouple before and during LPS induced hypothermia model. LPS was injected 1.5 h after the temperature recording. Data were presented as mean values based on three biologically independent mice ($n = 3$). Error bars were defined as standard deviation. **e** Ratiometric temperature imaging using LIBRA for temperature detection of mice cerebral vessels in control without LPS injection (left) and hypothermia after LPS injection (right). The regions labeled with white boxes in inferior cerebral veins (ICV), superior sagittal sinus (SSS), and the transverse sinus (TS) were used to measure the temperature values. **f** Statistical histograms of the temperature of mice cerebral vessels including ICV, SSS, and TS in control (denoted as Con ICV, Con SSS, and Con TS) and hypothermia (denoted as Hypo ICV, Hypo SSS, and Hypo TS) groups. Data were presented as mean values based on three biologically independent mice ($n = 3$). Error bars were defined as standard deviation.

of LIBRA@PEG was performed in solution and in the cerebral vessels of mouse with the same imaging parameters (Supplementary Fig. 32). In solution, the lifetime readout of LIBRA@PEG based on the high luminescence intensity was 68.4 μs (Supplementary Fig. 32a, d). However, in a mouse with intact scalp, the luminescence signals in cerebral vessels (Supplementary Fig. 32b) were reduced due to the absorption and scattering of biological tissues, leading to a corresponding lifetime readout deviation to 77.3 μs. Removal of the scalp mitigated the light extinction effect, enhancing luminescence intensity in the vessel and resulting in a lifetime reading of 67.0 μs (Supplementary Fig. 32c, f), which was close to the lifetime value in solution. These results suggested that the extinction effect of biological tissue (absorption and scattering) may partially interfere with lifetime-based readouts. The deviation might be reduced by increasing the integration time or sampling frequency during imaging. However, it still needs to prolong the signal collection time and sacrifice the temporal resolution that is a trade-off for the observation of the dynamic process in vivo.

In diagnosing diseases like hypothermia, where temperature is a critical indicator, rapid temperature detection with LIR is crucial for timely intervention to mitigate severe outcomes. The spatial resolution of nearly 200 μm offered by LIBRA's LIR thermometry also provides detailed insights into the hypothermia-affected cerebrovascular system in the brain, aiding in prognosis evaluation. This study introduces a minimally invasive tool for in vivo temperature measurement and elucidates a temperature response mechanism of lanthanide luminescence nanomaterials, which is of significance in the development of lanthanide luminescence nanothermometers in the future.

## Methods

### Synthesis β-NaErF₄:Yb/Y nanoparticles

One millimole of lanthanide chloride (0.8 mmol of ErCl₃, 0.2 mmol of YbCl₃ or YCl₃) combined with 6 ml of oleic acid (OA) and 15 ml of 1-octadecene (ODE) were added into a three-necked flask with a volume of 100 ml. Then, the precursors were degassed by stirring at room temperature for 30 min until no bubble was observed, followed by heating the mixture to 130 °C and degassing for 1 h to obtain a transparent solution. Thereafter, the solution was allowed to cool down to 60 °C. Four milliliters of methanol solution containing

2.5 mmol of NaOH and 4 mmol of $NH_4F$ were then gradually added to the flask in a dropwise fashion. The reaction system was opened to air for another 20 min at 70 °C to remove methanol. Subsequently, the solution was rapidly heated to a temperature of 300 °C and then sustained at this temperature for one hour in an atmosphere of nitrogen. The nanoparticles were separated using centrifugation at a force of 18,160 g/min for 15 min and underwent two rounds of washing with a 1:1 v/v mixture of ethanol and cyclohexane. The OA-capped nanoparticles were stored in 8 ml of cyclohexane for subsequent experiments.

## Synthesis β-$NaErF_4$:Yb@$NaYF_4$ core-shell nanoparticles

To establish the $NaYF_4$ shell layer, 0.5 mmol of $YCl_3$ was combined with 6 ml of OA and 12 ml of ODE and added into a three-necked flask with a volume of 100 ml. The following procedures are similar to the ones for preparing β-$NaErF_4$:Yb/Y nanoparticles. When $YCl_3$ was dissolved and the temperature of the system decreased to 60 °C, 0.5 mmol of β-$NaErF_4$:Yb/Y core nanoparticles dispersed in 3 ml of ODE was added and the system underwent degassing until no bubble was observed. Subsequently, 2 ml of methanol solution with 1.25 mmol of NaOH and 2 mmol of $NH_4F$ were introduced into the system dropwise. After the removal of methanol, the solution was rapidly heated to 300 °C and sustained for one hour under $N_2$. Core-shell nanoparticles were separated from solution by centrifugation (18,160 g/min) for 15 min and the precipitates were washed twice with a 1:1 v/v mixture of ethanol and cyclohexane. The products were stored in 4 ml of cyclohexane for the following experiments.

## Synthesis β-$NaErF_4$:Yb@$NaYF_4$:x% (x = 0, 20, 50, 100) Yb core-shell nanoparticles

The synthetic procedures of β-$NaErF_4$:Yb@$NaYF_4$:x% Yb core-shell nanoparticles were similar to β-$NaErF_4$:Yb@$NaYF_4$. In this synthesis, 0.25 mmol of lanthanide chloride (x% mol of $YbCl_3$ (x = 0, 20, 50, 100), and (100−x)% mol of $YCl_3$) were dissolved in OA and ODE. 0.25 mmol of β-$NaErF_4$:Yb nanoparticles were used as core. The obtained β-$NaErF_4$:Yb@$NaYF_4$:x% (x = 0, 20, 50, 100) core-shell nanoparticles were distributed in 4 ml of cyclohexane for subsequent procedures.

## Synthesis of core-shell nanoparticles with different shell thicknesses

The synthetic procedures of core-shell nanoparticles with different shell thicknesses were similar to β-$NaErF_4$:Yb@$NaYF_4$. For the synthesis of nanocomposites with inert shell, 0.25 mmol of β-$NaErF_4$:Yb nanoparticles were used as core, and 0.1 and 0.25 mmol of $YCl_3$ were used as precursors to form the inert $NaYF_4$ shell layers, resulting in the shell thickness of 1.6 and 3.7 nm, respectively. To reach an inert shell thickness of 12.7 nm, $NaErF_4$:20%Yb@$NaYF_4$ nanoparticles with a $NaYF_4$ shell thickness of 3.7 nm were used as core and 0.75 mmol $LuCl_3$ were used as precursors to coat a second shell layer of $NaLuF_4$, forming $NaErF_4$:20%Yb@$NaYF_4$@$NaLuF_4$ nanoparticles. For the synthesis of nanocomposites with active shell, $NaErF_4$:Yb@$NaYF_4$:Yb with varied shell thickness were prepared by controlling the molar ratio between lanthanide chloride (50% mol of $YCl_3$ and 50% mol of $YbCl_3$) and $NaErF_4$:Yb core nanoparticles. $NaErF_4$:Yb@$NaYF_4$:Yb nanoparticles with $NaYF_4$:Yb shell thickness of 0.52, 1.45, 3.06, and 5.91 nm were prepared corresponding to the molar ratio of 0.3:1, 0.6:1, 1:1, and 2:1, respectively.

## PEGylation of OA-capped LIBRA

Five milligrams of DSPE-PEG were added into 2 ml of dichloromethane with 20 s of sonication to get a homogeneous solution. Ten milligrams of nanoparticles kept in cyclohexane were added and then the mixture was sonicated for another 20 s. Thereafter, the solvent in the system was evaporated and the mixture of DSPE-PEG and nanoparticles formed a thin film on the wall of flask. Two milliliters of deionized water were added and the mixture was stirred for 2 h to obtain the DSPE-PEG modified LIBRA (LIBRA@PEG).

## Preparation of ligand-free nanoparticles

Ten milligrams of nanoparticles dispersed in cyclohexane were mixed with 1 ml of $NOBF_4$-saturated dichloromethane solution with gentle shaking. Precipitated nanoparticles were collected by centrifugation (12,611 g/min) for 8 min. The pellet was dispersed in deionized water or DMF through sonication.

## Relative quantum yield measurement of LIBRA

The relative quantum yield of LIBRA ($QY_{LIBRA}$) was determined by using IR-806 dye in DMF as a reference, which is reported to be 7.0%[72]. The quantum yield of LIBRA was measured based on the quantum yield of IR-806 dye in DMF, and the emission intensities and the absorptions of LIBRA and IR-806 in solutions. The emissions of LIBRA and IR-806 dye were excited by an 808 nm laser. The quantum yield of LIBRA was calculated according to Eq. 3[73,74],

$$QY_{LIBRA} = QY_{IR-806} \cdot \frac{E_{LIBRA}}{E_{IR-806}} \cdot \frac{A_{IR-806}}{A_{LIBRA}} \cdot \frac{I_{IR-806}}{I_{LIBRA}} \cdot \left(\frac{N_{LIBRA}}{N_{IR-806}}\right)^2 \quad (3)$$

where $QY_{LIBRA}$ and $QY_{IR-806}$ are the quantum yield of the LIBRA and IR-806, $E_{LIBRA}$ and $E_{IR-806}$ are the integrated emission intensity of LIBRA and IR-806, $A_{LIBRA}$ and $A_{IR-806}$ are the absorptions of LIBRA and IR-806 in the dispersion, $I_{LIBRA}$ and $I_{IR-806}$ are the relative excitation intensities of LIBRA and IR-806 and $N_{LIBRA}$ and $N_{IR-806}$ are the refractive indexes of the solvents for dispersing the samples. The quantum yield of LIBRA was 2.12 ± 0.08%.

## In vitro cytotoxicity of PEGylated LIBRA (LIBRA@PEG)

Cytotoxicity of LIBRA@PEG was tested using methyl thiazolyl tetrazolium (MTT) assay. Human embryonic kidney (HEK) 293 cells (Catalog Number: GNHu43) were provided by the Shanghai Institute of Biochemistry and Cell Biology, China. The cell line was tested with short tandem repeat (STR) for authentication. HEK 293 cells were first added into a 96-well plate with $1 \times 10^4$ cells in each well. The cells were then attached to the bottom of wells after incubating with 100 μl medium for 6 h under an atmosphere of 5% $CO_2$ at 37 °C. LIBRA@PEG dispersed in medium were supplemented to the cells with a series of concentrations at 0, 50, 100, 200, 400, 600, and 800 μg ml⁻¹ and then the cells were further cultured for 44 h. After that, twenty microliters of MTT solution (5 mg ml⁻¹) were dispensed into each well, followed by an additional incubation period of 4 h for the cells. After removal of the medium, 200 μl of dimethyl sulfoxide was added in each well. The plate was shaken gently for 10 min. The microplate reader was used to quantify the viability of cells by recording the absorption at wavelengths of 490 nm and 570 nm.

## Intracerebral temperature detecting through thermocouple

The animal procedures were approved by the Institutional Animal Care and Use Committee, ShanghaiTech University (ethical approval number: 20211115001). Seven-week-old female *Mus musculus* Balb/cA mice provided by Shanghai Jihui Laboratory Animal Breeding Co., Ltd (n = 3) were used for all experiments. Sex is not considered in this study because this study is to develop temperature imaging method and used cerebral vessels of mice to demonstrate the imaging spatial resolution, which did not study the possible biological difference among different sexes of the animal model. Before surgery, mice were anesthetized in a chamber with a continuous flow of 5% isoflurane mixed with 300 ml min⁻¹ oxygen until no response to external stimuli. Then, mice were placed in the prone position on the stereotaxic frame with nose covered by an anesthesia mask and ear bars were used to fix the head. A continuous flow of 1% isoflurane mixed with 200 ml min⁻¹

oxygen was used for the maintenance of anesthesia. The hairs on the head, neck, abdomen and leg of mice were removed using depilatory cream. Then, a semicircular cut was done on the scalp to expose the skull. A dental drill was used to create a small hole in the skull and the thermocouple was inserted into the motor cortex (mediolateral 0.5 mm, anteroposterior 1 mm, and dorsoventral 0.6 mm to bregma). Dental glue was used to immobilize the thermocouple and scalp. After the surgery, the anesthetic gas flow was turned off. Another thermocouple was fixed on the abdomen skin.

### Establishment of sepsis induced hypothermia model in mouse

To establish the hypothermia mouse model, 7-week-old female Balb/cA mice ($n = 3$) were intraperitoneally administered with lipopolysaccharide (LPS) at a dose of $50\,mg\,kg^{-1}$, which will induce sepsis (severe systemic inflammation) that can cause whole-body hypothermia (including brain) due to the endotoxin shock affecting the thermoregulatory of the body[75–77]. After LPS injection, a thermal camera was used to track the temperature of the scalp, and thermocouples implanted in the brain tissue and fixed on the abdomen skin were used to monitor the temperature changes of the brain and the body, respectively.

### NIR-II/III in vivo cerebral vascular imaging

NIR-II/III in vivo imaging system (Monet IGS-1000P, Suzhou NIR-Optics Co., Ltd., China) was used in mice cerebral vessel imaging. An 808 nm continuous laser with a power density of $120\,mW\,cm^{-2}$ worked as an excitation source and a 900 nm long-pass filter and 1400 nm long-pass filter were put in front of the NIR-II camera to acquire luminescence imaging at varied wavelengths. In vivo near-infrared imaging experiments were carried out 2 min after LIBRA@PEG (200 μl, $40\,mg\,ml^{-1}$ per mouse) was injected into 7-week-old female Balb/cA mice ($n = 3$) through the tail vein.

### Light propagation model establishment for the calibration of luminescence intensity ratio-based temperature measurement in vivo

To give the accurate relationship between the luminescence intensity ratio (LIR) and temperature in the biological tissue, the light propagation model that describes the luminescence intensity in the medium with absorption and scattering coefficients (e.g., biological tissue) was established based on the photon diffusion equation with the Green's function as a solution. The luminescence intensities of the emissions at 980 and 1550 nm of LIBRA were expressed by the model and the relationship of luminescence intensity ratio and temperature was calibrated with Eq. 4,

$$T = \frac{\alpha}{\gamma} R_I + \beta \tag{4}$$

where $R_I$ is the luminescence intensity ratio ($I_{1550}/I_{980}$); $T$ is temperature; $\alpha$ and $\beta$ are determined by the temperature response of LIBRA without medium; $\gamma$ is a correction term that is expressed as Eq. 5,

$$\gamma = \frac{D(\lambda_{980})}{D(\lambda_{1550})} e^{(\mu_{eff}(\lambda_{980}) - \mu_{eff}(\lambda_{1550})) \cdot d} \tag{5}$$

in which $D(\lambda)$ is the diffusion coefficient and $\mu_{eff}(\lambda)$ is the effective attenuation coefficient of the emissions at 980 and 1550 nm and $d$ is the measured tissue depth. The luminescence intensities of the two emissions after light propagation were experimentally measured by the NIR-II/III bioimaging and the luminescence intensity ratio was brought into the model so that the temperature can be determined. Detailed derivations and validation of the model were shown in Supplementary Notes 1.

### Near-infrared lifetime imaging

The lifetime imaging was performed in a homemade imaging system. An 808 nm laser diode coupled with optical fiber was used for luminescence excitation operating at 1 Hz with a duty cycle of 20% at power density of $50\,mW\,cm^{-2}$ in average. Luminescence images beyond 1500 nm were captured using an InGaAs camera that received a triggering signal at a precise delay time to be synchronized with the excitation source. The camera was exposing prior to the subsequent laser pulse's arrival. A set of time delays were used to acquire different sections in the luminescence decay profile. The lifetime value of the pixels in the luminescence lifetime image was fitted using the equation, $I_t = I_0 \times e^{-t/\tau}$, where $t$ stands for the delay time, $I_0$ represents the initial luminescence intensity at $t = 0$, and $\tau$ stands for the lifetime of the nanoparticles.

### Reporting summary

Further information on research design is available in the Nature Portfolio Reporting Summary linked to this article.

## Data availability

All the data generated and analyzed from this study are presented in the article and its Supplementary Information files. The data are also available from the corresponding author upon request. Source data are provided with this paper.

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

## Acknowledgements

This work is supported by National Natural Science Foundation of China (82001945 (X.Z.), 62105205 (W.R.)), Shanghai Pujiang Program (20PJ1410700 (X.Z.)), the starting grant of ShanghaiTech University (X.Z. and W.R.), and Shanghai Clinical Research and Trial Center (X.Z.). We sincerely appreciate Prof. Chunyan Li, Prof. Yejun Zhang, and Dr. Feng Wu for the precious help in luminescence lifetime imaging and Prof. Wei Feng and Mr. Jiaming Ke for the generous help in the measurement of luminescence lifetime.

## Author contributions

Yukai Wu, Xingjun Zhu, and Fuyou Li proposed the concept. Yukai Wu and Fang Li designed the methodology, conducted experiments, and established the model for the study. Yanan Wu, Liangtao Gu, and Yukun Qi performed light propagation measurement and computing. Hao Wang and Lingkai Meng performed material synthesis. Fang Li, Hao Wang, and Yingjie Chai contributed to luminescent life measurements and data analysis. Jieying Zhang, Na Kong, Qian Hu, and Zhenyu Xing contributed to material characterizations. The manuscript was collectively written by all authors, and all authors have given their approval for the final version. The manuscript was supervised by Wuwei Ren, Fuyou Li, and Xingjun Zhu.

## Competing interests

The authors declare no competing interests.
