## [Peer Review File · Nature Communications]

Lanthanide luminescence nanothermometer with working wavelength beyond 1500 nm for cerebrovascular temperature imaging in vivoEditorial Note: Parts of this Peer Review File have been redacted as indicated to remove third-party material where no permission to publish could be obtained.

REVIEWER COMMENTS

Reviewer #1 (Remarks to the Author):

In this work authors introduce a new type of luminescence nanothermometer that is used, as a proof of concept, to measure temperatures of the cardiovascular system of the brain. From my point of view the novel point of this work is the design and development of the nanothermometer that is based on the use of the water induced quenching/enhancement of luminescence to provide thermal sensitivity to a simple(smart) core/shell structure. The thermal sensitivities reported are good when compared to those of other lanthanide doped nanomaterials but are significantly smaller than those reported for other nanothermometers also operating in the second/third biological window. The final proof of concept provided by authors is nice as it attempts to measure temperature in the brain. Nevertheless I find that the paper fails to discuss in detail the reliability of their thermal probe. It is based on the measurement of the ratio between the emitted intensities at 1500 and 980 nm generated by different ions by from the same nanoparticle. This approach has two problems. First it is well known that the detected emitted intensity at 1500 nm depends a lot on the thickness of the aqueous medium between the excitation beam and the detection system (water absorption is quite strong at this wavelength). Thus, according to this approach, the thermal readout would be affected by the depth within a fluid or within a tissue (also contains water) at which nanothermometers are located. So I have serious doubts about the reliability of the thermal readout of this approach because this fact. At the same time, may be I am wrong, it seems that the sensitivity could depend strongly on the water content in the medium. It is clear that authors known this possibility because they include in main text and SI some calibration curves obtained in different media but they do not measure the calibration curve in blood. Note that in blood the formation of a protein corona is expected to occur and the transmittance of blood in the infrared is expected to be affected by the oxygenation level. So it is quite possible that calibration would be different in blood. Also the presence of bias between temperature and blood conditions (such as

oxygenation level) should be discarded.

The two points mentioned above are critical. But I think that authors can explore how relevant they are by exploring the calibration curve in blood and at different depths in tissues and liquid media. In addition, authors also report that they nanostructures are capable of providing a thermal readout by the use of lifetime. Thus it seems that this provides a way to check for the reliability of their measurements. Note that when a single nanothermometer is capable of providing thermal readouts by the analysis of different parameters this opens a way of check for the correctness and reliability of thermal readouts. Also it is nowadays consider that lifetime based thermal measurements are the most trustable ones. So I think authors should measure the thermal images by analyzing lifetime instead of intensity rations to check for the reliability of their results.

Reviewer #2 (Remarks to the Author):

The review of a manuscript entitled “Lanthanide luminescence nanothermometer with working wavelength beyond 1500 nm for cerebrovascular temperature imaging in vivo” by Xingjun Zhu et al..

The articles present a novel concept for thermometry, where the ratio of 1550nm emission from core Er³⁺ ions is compared to 980 nm emission of Yb³⁺ (quenched by H₂O), which shows temperature dependence. The intensity of 1550nm emission (of Er³⁺) drops vs temperature more rapidly as compared to 980 nm (from Yb³⁺), where temperature dependent quenching of Yb decrease the population of 4I_{13/2} level (the source of 1550nm emission). The minimally invasive, remote and high-resolution temperature (bio)imaging in vivo (of brain) is an important field of applied science.

The figures are readable and clear, references are up-to date, but some reviews on temperature imaging exist which could be cited and the results of this work could be confronted with current state-of-the-art a little bit more (other temperature reading methods, other nanothermometers etc.). The message and idea is relatively clear and interesting. The methodology is sound and clear.

The work generally support the conclusions and claims, but some additional evidence are needed - Not only scattering is an issue with in vivo imaging but the sample spectroscopy as well (See discussion on Jaque et al, and Bednarkiewicz et al. papers). Even though the authors calibrated their thermometers with tissue sample, many more artifacts may occur (water presence, NPs presence/concentration/aggregation). The temperature readout may also be hampered by variations in the concentration of nanothermometers in limited volume. Thus, to confirm the reliability the authors must study the readout accuracy for various concentrations (up to very high - aggregate like samples) of nanoparticles. The concentration and aggregates has shown the variations is spectral properties as compared to individual (or diluted) luminescent nanoparticle. Moreover, the properties of nanothermometers with PEG should be studied with LPS against the possible aggregation and change in spectral properties. This discussion and analysis of possible artifacts are critical for in vivo thermometry and reliable conversion of the LIR to temperature.

I would not call the energy transfer from Yb do O-H as LRET. Typically this is quenching by O-H groups or overtones of water molecule vibrations.

Luminescence nanothermometry is not 'absolutely' noninvasive – the nanoparticles must be inserted and their aggregates or dissolution may potentially bring side and harmful effects. These potential toxicity issues should at least briefly be mentioned with respective literature / reviews references.

Other nanothermometers has been shown to successively detect temperature in vivo, which are based on quantum dots (e.g. Ag2S). This should supplement the state-of-the-art discussion.

I do not understand what authors mean by 'bidirectional'.

"LIBRA with NaYF₄:50%Yb shell showed a 23-fold enhancement of the emission at 1550 nm compared to Core-LnNP (Fig. 2i)." - the mechanisms of 1550 nm emission enhancement is not clearly explained in the manuscript. Having Yb doped shell should depopulate 4I_{11/2} level of Er, which is feeding 4I_{13/2} level. So, is the 23-fold enhancement 'absolute' or 'relative' to the 980 nm emission band?

In general, often, no in-depth discussion on mechanism / phenomena behind is provided. It is not enough to describe what one can see on the graphs. E.g. why the lifetimes (fig.4d) get shorter with temperature? Why both passive and active shells have similar relative sensitivities (Fig.4g) ? Why DMF quench I980 ? What are the black up-arrows in Er (Fig.1) ? “the intensity of 1550 nm emission increased gradually and the intensity at 980 nm decreased with temperature” – but what is the mechanism behind? Why was a linear model proposed to model the experimental data set (see ref.12]) ?

Why the emission lifetime of $4I_{13/2}$ is so short? I would expect it to be significantly longer.

Some comments to figures:

There is no error estimation of e.g. Fig.3, taking into account errors in establishing Er and Yb emission. Why there is no temperature quenching at 1550 in DMF (fig.4b) when slight Yb quenching is observed in DMF ?

Fig.6. Shouldn't LPS (and sepsis) induce fever (hyperthermia) rather than hypothermia? Can the NP penetrate through blood-brain barrier after I.V. injection? While the 808 nm is good for bio, the 980nm emission from Yb overlap with water, thus on the path between the nanothermometers to detector, the amount/length/type of tissue may disrupt the 1550/980 ratio and may provide misleading data based on calibration data. Taking into account that no one knows the depth the emission comes from the calibration, the temperatures presented require more in-depth critical analysis and discussion.

Reviewer #3 (Remarks to the Author):

In this work, the authors proposed a bidirectional-responsive ratiometric nanothermometer (abbreviated as LIBRA) based on NaErF₄:Yb@NaYF₄:Yb nanoprobe for temperature imaging in vivo. The luminescence resonant energy transfer between lanthanide ions and water molecules resulted in temperature-sensitive emissions at 1550 nm and 980 nm, the emission intensity ratio of which can be applied for temperature detection. Based on the NaErF₄:Yb@NaYF₄:Yb nanoprobe, noninvasive temperature imaging of mouse

cerebrovascular system was realized with a thermometric imaging resolution of $\sim 200 \mu\text{m}$. It is a systematic study and the topic itself may be of great interest for the researchers in this field. However, some points of the study should be further clarified before publication.

1. The authors mentioned that few studies reported the temperature monitoring of micron-sized patterns in vivo because the luminescence intensity of NIR-II/III nanothermometer was not optimized. What is the NIR quantum yield of the proposed $\text{NaErF}_4:\text{Yb}@/\text{NaYF}_4:\text{Yb}$ nanoprobles?

2. Supplementary Figure 4 shows the DLS measurements of DSPE-PEG modified LIBRA in deionized water, PBS and 37 °C FBS solution, which indicates the hydrodynamic diameters of DSPE-PEG modified LIBRA are much larger than that of the OA-capped LIBRA.

Transmission electron microscopy images of the DSPE-PEG modified LIBRA in different solutions are suggested to be provided.

3. Figure 3d displays the relative thermal sensitivity of LIBRA@PEG. The temperature uncertainty (δT) as another important thermal parameter is suggested to be provided.

4. When LIBRA was dispersed in DMF, the intensity of 980-nm emission decreased while the 1550-nm emission was almost unchanged with increasing temperature. Why? Please explain it.

Responses to the Reviewers' Reports Manuscript (NCOMMS-22-37056)

Reviewer #1 (Remarks to the Author):

Comment:

In this work authors introduce a new type of luminescence nanothermometer that is used, as a proof of concept, to measure temperatures of the cardiovascular system of the brain. From my point of view the novel point of this work is the design and development of the nanothermometer that is based on the use of the water induced quenching/enhancement of luminescence to provide thermal sensitivity to a simple(smart) core/shell structure. The thermal sensitivities reported are good when compared to those of other lanthanide doped nanomaterials but are significantly smaller than those reported for other nanothermometers also operating in the second/third biological window. The final proof of concept provided by authors is nice as it attempts to measure temperature in the brain.

Nevertheless I find that the paper fails to discuss in detail the reliability of their thermal probe. It is based on the measurement of the ratio between the emitted intensities at 1500 and 980 nm generated by different ions by from the same nanoparticle. This approach has two problems.

First it is well known that the detected emitted intensity at 1500 nm depends a lot on the thickness of the aqueous medium between the excitation beam and the detection system (water absorption is quite strong at this wavelength). Thus, according to this approach, the thermal readout would be affected by the depth within a fluid or within

a tissue (also contains water) at which nanothermometers are located. So I have serious doubts about the reliability of the thermal readout of this approach because of this fact.

At the same time, maybe I am wrong, it seems that the sensitivity could depend strongly on the water content in the medium. It is clear that authors know this possibility because they include in main text and SI some calibration curves obtained in different media but they do not measure the calibration curve in blood. Note that in blood the formation of a protein corona is expected to occur and the transmittance of blood in the infrared is expected to be affected by the oxygenation level. So it is quite possible that calibration would be different in blood. Also the presence of bias between temperature and blood conditions (such as oxygenation level) should be discarded. The two points mentioned above are critical. But I think that authors can explore how relevant they are by exploring the calibration curve in blood and at different depths in tissues and liquid media.

In addition, authors also report that their nanostructures are capable of providing a thermal readout by the use of lifetime. Thus it seems that this provides a way to check for the reliability of their measurements. Note that when a single nanothermometer is capable of providing thermal readouts by the analysis of different parameters, this opens a way to check for the correctness and reliability of thermal readouts. Also it is nowadays considered that lifetime based thermal measurements are the most trustworthy ones. So I think authors should measure the thermal images by analyzing lifetime instead of intensity ratios to check for the reliability of their results.

Response: We appreciate the reviewer for the approval of our work and the detailed advices. The main contribution and innovation of this work is to provide a visualized temperature sensing method with high spatial resolution based on near infrared emissive nanomaterials for detecting the temperature change of micron-sized pattern represented by cerebral blood vessels in the living body. To achieve this goal, the use of luminescence intensity ratio (LIR) is necessary and to some extent indispensable because LIR has the merits of fast detection speed, and steady-state excitation and signal acquisition with sufficient signal intensity for high spatial resolution imaging. In the case of the temperature imaging of cerebral blood vessel, it is important to conduct the thermometry within a time frame as short as possible because the clearance of the luminescent imaging agent in the circulatory system due to metabolism process will result in the decrease of luminescence intensity as time goes, leading to the deviation of temperature readout. LIBRA presented in this work showed minimal variation of optical signal *in vivo* over 120 s (Fig. 5b) and was designed with the ratiometric strategy (quick temperature detection), making it an ideal candidate for detecting the temperature of cerebral vessels.

Fig. 5 | b Time-dependent quantitative cerebral vessel luminescence signal of control mouse and hypothermia mouse.

The comments of the referee are valuable and inspired us to take more thinking and analysis of the factors influencing the accuracy of luminescence probe based nanothermometry. Lifetime decoding, which is based on the periodic transient-state excitation and the acquisition of the time-dependent decay of luminescence intensity signals, requires a relatively long time to accumulate sufficient luminescence intensity signals for precise lifetime readout. Since the detecting signal of lifetime decoding also depends on the luminescence intensity, lifetime-based nanothermometry due to its relatively slow detection speed is not suitable for temperature detection of cerebral vessels as the metabolism of luminescent probe in the blood circulatory system will decrease the luminescence intensity, which will affect the accuracy of lifetime decoding and thus bring about additional deviations (Supplementary Fig. 34).

Supplementary Fig. 34 Scheme demonstrating the additional intensity decrement by metabolism of nanoparticle in blood circulatory system.

As a simplified demonstration, for the collection of luminescence intensity of the imaging agent in the blood circulatory system *in vivo* to decode lifetime, the change rate of luminescence intensity can be expressed by the following formula.

$$\frac{dI}{dt} = -(k_1 + k_2)I$$

where $\frac{dI}{dt}$ indicates the luminescence decrement per unit time and it consists of two parts, which are contributed by the clearance rate (k_1) of the blood circulatory system and the decay rate of excited state energy levels (k_2).

After integration, the relationship between luminescence intensity and time is obtained as the following equation,

$$I = I_0 e^{-(k_1+k_2)t} = I_0 e^{-\frac{t}{(k_1+k_2)^{-1}}}$$

$$\tau_a = (k_1 + k_2)^{-1}$$

Compared to the intrinsic luminescence lifetime of imaging agent ($\tau_i = k_2^{-1}$), which equals to the reciprocal of the decay rate of excited state energy levels, the decoded apparent lifetime ($\tau_a = (k_1 + k_2)^{-1}$) of the imaging agent in the blood circulatory system of the living body contains the decrease of luminescence intensity due to the imaging

agent clearance, and thus deviates from the intrinsic luminescence lifetime, which affects the accuracy of thermometry.

Furthermore, the lifetime-based reading derived from luminescence intensity can also be influenced by the absorption and scattering of bio-tissue at a certain extent. To illustrate this, luminescence lifetime imaging using the 1550 nm emission of LIBRA as an example was performed in solution and in the cerebral vessel of mouse with the same imaging parameters (Supplementary Fig. 35). As the lifetime change of LIBRA is too little upon temperature (Supplementary Fig. 16), the lifetime reading of LIBRA was conducted instead of lifetime thermometry. The lifetime readout of LIBRA in solution based on the high luminescence intensity was 68.4 μs (Supplementary Fig. 35a and d). However, in a mouse with intact scalp, the luminescence intensity in cerebral vessel (Supplementary Fig. 35b) was reduced due to absorption and scattering of bio-tissue resulting in a corresponding lifetime readout deviation to 77.3 μs . Once the scalp was removed, the luminescence intensity in cerebral vessel improved due to the reduced extinction effect (Supplementary Fig. 35c), and the corresponding lifetime readout was 67.0 μs (Supplementary Fig. 35f) which is close to the lifetime value in solution (Supplementary Fig. 35d). The results suggest that the extinction effect of bio-tissue (absorption and scattering) may partially interfere with lifetime-based readouts.

Hence, the accuracy of lifetime-based temperature reading method is also affected by the factors existing in vivo that disturb the collection of luminescence intensity signals. The deviation might be reduced by increasing the integration time or sampling frequency during imaging while it still needs to prolong the signal collection time and

sacrifice the temporal resolution that is a trade-off for the observation of the dynamic process in vivo. We understand and appreciate the referee for providing the suggestions to improve our work. As both luminescence intensity ratio and luminescence lifetime for nanothermometry have their own applicability, it is necessary to take multiple parameters (spatial resolution, temporal resolution, accuracy and etc.) into account during the thermometry of the different biological models, which can achieve the advances in different aspects and create the opportunity for the solution of comprehensive application of nanothermometry.

Supplementary Fig. 35 Luminescence intensity images beyond 1500 nm of LIBRA in the centrifuge tube (a), cerebral vessel with intact scalp (b) and one with sheared scalp (c). Corresponding luminescence lifetime images beyond 1500 nm of LIBRA were shown in d, e and f. The mice were euthanized after LIBRA was injected through caudal vein.

As suggested by the referee, the calibration curves of LIR of LIBRA *versus* temperature are collected in deionized water (H₂O), phosphate-buffered saline (PBS), normal saline (0.9 % NaCl saline), fetal bovine serum (FBS) and blood from vein

(Supplementary Fig. 32). Although the profiles exhibit different fitting parameters in various dispersion environment, the calibration curve should be measured in a setup that is identical to the specific biological condition during in vivo temperature detection to better discriminate the temperature change in vivo. Hence, we established the setup simulating the condition of cerebral vessel in mouse for calibration curve measurement (Supplementary Fig. 31). The blood vessels observed in this work (inferior cerebral veins (ICV), the superior sagittal sinus (SSS), superficial veins (SV), and the transverse sinus (TS)) are vein vessels distributing on the top of cortex. The biological tissues including scalp, periosteum, skull and etc. covering the veins have identical thicknesses, which are measured in this work to be 0.69 ± 0.02 mm, so it is reasonable to conclude that these veins have similar tissue depth and the condition of extinction coefficient. In the setup for calibration curve measurement, the blood from vein is mixed with LIBRA and is filled into the capillary tube with diameter of 200 μm to simulate the cerebral blood vessels observed in the in vivo demonstration. The intact tissues above mouse brain including scalp, periosteum, skull and etc. are used to cover the capillary tube to build the tissue scattering and absorption parameters (Supplementary Fig. 31). The calibration curve obtained from the setup was shown in Supplementary Fig. 33. For in vivo demonstration, normal mouse and hypothermia mouse injected with LIBRA exhibited different LIR in the cerebral blood vessels based on the near infrared luminescence imaging, which confirms that LIBRA can discriminate the temperature difference in the living body for hypothermia diagnosis (Figure 6e and 6f). By using the calibration curve for in vivo thermometry to calculate the temperature difference of

the cerebral blood vessels between normal mouse and hypothermia mouse, it showed that the temperature difference obtained from the LIR of LIBRA is 3.94 °C. The temperature decrement of brain tissue measured by the transcranial implanted thermocouple is 8.5 °C (Figure 6d). These results imply that the relative temperature change detected by LIBRA can reflect the hypothermia symptom in cerebral vascular system and thus explains one of the reasons for the temperature change of the brain tissue. The fast diagnosis of hypothermia is important to guide the treatment in time and the spatial resolution based on the LIR thermometry of LIBRA will also provide clear information of the hypothermia affected brain regions to evaluate the prognosis. The concept presented in this work aiming at the high spatial resolution and fast temperature detection points to the biological application of the nanothermometer, of which the progress is the advance in this field.

[FIGURE REDACTED]

Supplementary Fig. 31 Illustration of the instrument for establishing *in vivo*-like calibrate curve.

Supplementary Fig. 32. Calibration curves of LIR of LIBRA *versus* temperature are in phosphate-buffer solution (PBS), normal saline, deionized water, fetal bovine serum (FBS) and blood from vein.

Supplementary Fig. 33. Temperature calibrate curve of LIBRA for in vivo thermometry based on the set up in Supplementary Fig. 31.

Reviewer #2 (Remarks to the Author):

The review of a manuscript entitled Lanthanide luminescence nanothermometer with working wavelength beyond 1500 nm for cerebrovascular temperature imaging in vivo by Xingjun Zhu et al..

The articles present a novel concept for thermometry, where the ratio of 1550 nm emission from core Er^{3+} ions is compared to 980 nm emission of Yb^{3+} (quenched by H_2O), which shows temperature dependence. The intensity of 1550nm emission (of Er^{3+}) drops vs temperature more rapidly as compared to 980 nm (from Yb^{3+}), where temperature dependent quenching of Yb decrease the population of $^4I_{13/2}$ level (the source of 1550nm emission). The minimally invasive, remote and high-resolution temperature (bio)imaging in vivo (of brain) is an important field of applied science.

Comments 1:

The figures are readable and clear, references are up-to date, but some reviews on temperature imaging exist which could be cited and the results of this work could be confronted with current state-of-the-art a little bit more (other temperature reading methods, other nanothermometers etc.). The message and idea is relatively clear and interesting. The methodology is sound and clear.

Response: We appreciate the reviewer for the approval of our work and valuable suggestions. Additional review papers discussing about the temperature imaging and luminescence thermometry are cited in the revised manuscript.¹⁻⁶ Temperature reading methods include the imaging-based thermometry, contact probes and luminescence nanothermometers. For imaging-based thermometry, infrared thermal imaging can only detect the temperature of the surface. Magnetic resonance spectroscopy or photoacoustic imaging is still limited by the imaging spatial resolution and/or imaging speed. For contact probing methods, the high invasiveness is the main concern when it is necessary to detect the temperature inside the tissue. Luminescence thermometry by using nanomaterials such as lanthanide nanocomposite, quantum dots, nanodiamond and fluorescence polymer nanoparticles is a promising thermometric strategy for fast temperature reading and the imaging resolution depends on the emission wavelength. The luminescence nanothermometers with thermometric working wavelengths longer than 1350 nm (also known as NIR-III), which can bring about ideal temperature imaging resolution are still rare. As a result, it is urgent and necessary to develop long working wavelength nanothermometer so that the temperature detection of fine

biological structures can be achieved. In our work, lanthanide luminescence nanocomposite with the working wavelength beyond 1500 nm (NIR-III) is designed for the temperature imaging of the micron-sized biological structure in the living body. With the long wavelength temperature sensitive emission, the spatial resolution of temperature imaging can be improved into the level of 200 μm , which is of great significance in elucidating the microscopic temperature and its corresponding biological effect in vivo with an intuitive fashion.

As shown in Supplementary Table 2, temperature reading techniques for biomedical applications including imaging-based thermometry (infrared thermal imaging, magnetic resonance spectroscopy and photoacoustic imaging), luminescent nanothermometers and contact probe are summarized for comparison.

Supplementary Table 2 Temperature reading methods used for biomedical application.

Temperature imaging techniques	Biological model	Stimuli	Surface or interior	Real time or not?	Reference
Infrared thermal imaging	Mouse tumor	Photothermal therapy	Surface	Yes	10
Magnetic resonance spectroscopy	Human brain	-	Interior	No	11
Magnetic resonance	Human brain	Acute ischemic stroke	Interior	No	12

spectroscopy					
Magnetic resonance spectroscopy	Human brain	Ice slurry ingestion	Interior	No	13
Photoacoustic imaging	Human prostate	Cryotherapy	Interior	Yes	14
Photoacoustic imaging	Ex-vivo bovine tissue	High-intensity focused ultrasound therapy	Interior	Yes	15
Nanothermometer	Mouse limb vessel	Heating pad	Interior	Yes	2
Nanothermometer	Mouse hyperthermia liver	Laser heating	Interior	Yes	16
Nanothermometer	Mouse Subcutaneous tissue	Laser heating	Interior	Yes	8
Optical fiber	Mouse brain	Freely behaving mouse	Interior	Yes	17,18
Nanothermometer	Mouse brain	Hypothermia induced by LPS	Interior	Yes	This work

Comments 2

The work generally support the conclusions and claims, but some additional evidence are needed. Not only scattering is an issue with in vivo imaging but the sample spectroscopy as well (See discussion on Jaque et al, and Bednarkiewicz et al. papers). Even though the authors calibrated their thermometers with tissue sample, many more artifacts may occur (water presence, NPs presence/concentration/aggregation). The temperature readout may also be hampered by variations in the concentration of

nanothermometers in limited volume. Thus, to confirm the reliability the authors must study the readout accuracy for various concentrations (up to very high - aggregate like samples) of nanoparticles. The concentration and aggregates has shown the variations is spectral properties as compared to individual (or diluted) luminescent nanoparticle.

Response: Thanks for your advice. As shown in Supplementary Fig. 8, there was no change in spectral properties including peak position and peak shape when concentration changed from 4 to 10 mg/ml. The ratio of I_{1550} and I_{980} (LIR), which implied the thermal readout, kept quite stable upon different concentrations. As a result, the concentration and aggregates do not affect the thermal readout of LIBRA. Besides, the temperature response behavior was dependent on the dispersed environment (Figure 4b and Supplementary Fig. 9). In the revised manuscript, the relationship between concentration and thermal readout is discussed in “Results” section: “Temperature response behavior and mechanism of LIBRA.”:

“Additionally, LIR kept stable upon different concentrations of LIBRA in the aqueous dispersion, indicating that the concentration and potential aggregates do not affect the thermal readout (Supplementary Fig. 8).”

Supplementary Fig. 8 (a) Luminescence spectra of LIBRA of varied concentrations in aqueous environment at room temperature. (b) Luminescence intensity ratio (I_{1550}/I_{980}) of LIBRA in varied concentrations of aqueous solutions. Error bars, defined as s.d., were based on three measurements by spectrometry.

Fig. 4 | b. NIR luminescence spectra of LIBRA solid powder and those dispersed in water solution at 10, 50 and 90 °C.

Supplementary Fig. 9 NIR luminescence spectra of LIBRA dispersed in DMF at 10, 50 and 90 °C.

Comments 3:

Moreover, the properties of nanothermometers with PEG should be studied with LPS against the possible aggregation and change in spectral properties. This discussion and analysis of possible artifacts are critical for in vivo thermometry and reliable conversion of the LIR to temperature.

Response: We appreciate the advice. LPS was injected intraperitoneally to induce hypothermia to Balb/c mice while the nanothermometers with PEG were given by caudal vein injection. The administration pathways of LPS and nanothermometers in vivo are separated so that LPS would not interact with nanothermometers directly to change their spectral properties. To avoid misunderstanding of the LPS administration, the establishment of hypothermia mouse model is supplemented in the revised manuscript in the “Methods” section:

“Establishment of sepsis induced hypothermia model in mouse.

To establish sepsis induced hypothermia mouse model, female Balb/c mice were intraperitoneally administrated with lipopolysaccharide (LPS) at a dose of 50 mg kg⁻¹, which will induce sepsis (severe systemic inflammation) that can cause whole body hypothermia (including brain) due to the endotoxin shock affecting the thermoregulatory of the body⁷¹⁻⁷³. After LPS injection, thermal camera was used to track the temperature of scalp and thermocouples implanted in the brain tissue and fixed on the abdomen skin were used to monitor the temperature changes of the brain and the body, respectively.”

Comments 4:

I would not call the energy transfer from Yb to O-H as LRET. Typically this is quenching by O-H groups or overtones of water molecule vibrations.

Response: We appreciate this valuable suggestion. As shown in Fig. 1, the excited energy level of lanthanide ions was quenched to lower lying energy level or ground state by the stretch vibration and overtones of water molecules. The lower lying energy level facilitated by water quenching effect could assist the downshifting emission. In the revised manuscript, we modified the description of this energy transfer process as environment quenching assisted downshifting (EQAD), which is more suitable than LRET. The related descriptions are amended in the revised manuscript.

[FIGURE REDACTED]

Fig. 1 | Schematic of NIR-II/III emissive lanthanide ions doped nanocomposite as bidirectional ratiometric nanothermometer (NaErF₄:Yb@NaYF₄:Yb, abbreviated as LIBRA) for cerebral vessel temperature detection.

Comments 5:

Luminescence nanothermometry is not absolutely noninvasive. the nanoparticles must be inserted and their aggregates or dissolution may potentially bring side and harmful effects. These potential toxicity issues should at least briefly be mentioned with respective literature / reviews references.

Response: We appreciate this precious comment. To make the descriptions more accurate, the word 'noninvasive' were replaced by 'minimally invasive' in the revised

manuscript. The biosafety and potential toxicity of nanoparticles are discussed in the revised manuscript with respect to some reviews regarding this aspect^{7, 8} Please see the related description in the “Results” section, “NIR-III cerebral vessel imaging and temperature detection using LIBRA.”:

“The biosafety and potential toxicity of nanoparticles are related to the interaction of biological system and the physicochemical properties of nanoparticles such as chemical composition, dissolution, surface state and physical agglomeration and so on. For lanthanide nanomaterials, it is reported that lanthanide ions are not highly toxic^{67, 68}. With the modification of surface ligand like polyethylene glycol (PEG) and application of low dissolution rate matrix materials (sodium lanthanide fluoride), the release of lanthanide ions and the particle agglomeration are suppressed and the protein absorption on the nanoparticle surface can be reduced to alleviate immune response.”

“For the biological application, the toxicity of LIBRA was evaluated in vitro and in vivo. The cytotoxicity was assessed by using methyl thiazolyl tetrazolium (MTT) assay. The results as illustrated by Supplementary Fig. 22 indicated that LIBRA@PEG did not have obvious toxicity even under a high incubating concentration of 800 $\mu\text{g ml}^{-1}$ with the viability of ~85%. Besides, the main organs (heart, liver, spleen, lung and kidney) were collected from the mouse 7 days post intravenous injection with LIBRA and then for hematoxylin and eosin (H&E)-staining analysis. As shown in Supplementary Fig. 23, compared to control group without LIBRA injection, no noticeable lesion or injuries were observed for mouse treated with LIBRA, indicating

that the administration of LIBRA into mouse model would not induce significant toxicity to living animals.”

Supplementary Fig. 22 Methyl thiazolyl tetrazolium (MTT) assay of human embryonic kidney (HEK) 293 cell line incubated with LIBRA (0-800 µg ml⁻¹).

Supplementary Fig. 23 H&E-stained histological sections of the main organs (heart, liver, spleen, lung, kidney) from healthy mice (control) and the mice injected with LIBRA (experiment, 7 days after injection).

Comments 6:

Other nanothermometers has been shown to successively detect temperature in vivo, which are based on quantum dots (e.g. Ag₂S). This should supplement the state-of-the-art discussion.

Response: Thanks for your valuable advice. Some beautiful works regarding temperature detection in vivo based on Ag₂S and other quantum dots has been reported these years. The discussion of these works has been supplemented in the “Introduction” section. Please also see the content below,

“Recently near infrared emitting quantum dots such as Ag₂S and PbS have also been developed as nanothermometers for the temperature detection in vivo to identify the inflammation lesions or monitor the temperature increase during the thermal ablation of tumor⁹⁻¹¹ (Supplementary Table 1).”

We have integrated them into the Supplementary Table 1.

Supplementary Table 1 Parameters of the representative nanothermometer for bio-application reported in scientific papers. λ_{Ex} , λ_{Em} , S_r^{max} correspond to excitation wavelength, working wavelength and the highest relative sensitivity, respectively.

Nanothermometer	$\lambda_{Ex}/\lambda_{Em}$ (nm)	Subject detected	Sensing strategy	S_r^{max} (% K ⁻¹)	Imaging or not?	Spatial resolution	Reference
Ag ₂ S	808/1200	Inflamed liver	Lifetime	3	No	-	1
NaNdF ₄ :Yb@CaF ₂	785/1000	Leg vessel	Lifetime	~2 (298 K)	Yes	-	2
NaYF ₄ @NaYF ₄ :Yb ³⁺ , Nd ³⁺ @CaF ₂	800/980	Inflamed tissue	Lifetime	1.4 (283 K)	Yes	-	3
NaYF ₄ :	980/525,	Mitochon	Ratio	3.2	No	-	4

Yb ³⁺ , Er ³⁺	545	ndria in Hela cells		(305 K)			
PbS- NaYbF ₄ :T m@NaYF ₄ :Yb@Na YF ₄ :Nd	865/810	Tumor tissue	Ratio	5.6 (318 K)	Yes	-	5
Ag ₂ S	808/1200	Hyperthe rmia liver	Single peak intensity	3.9 (293 K)	Yes		6
Ag ₂ S	808/1200	Brain tissue	Single peak intensity	3	No	-	7
Er- Yb@Yb- Tm LaF ₃	690/1000 , 1200	Subcutan eous tissue	Ratio	5	Yes	-	8
PbS/CdS/ ZnS	808/1270	Tumor	Single peak intensity	-	No	-	9
NaErF ₄ :Y b@NaYF ₄ :Yb	808/1550 , 980	Cerebral vascular	Ratio	1.9	Yes	~200 μm	This work

Comments 7:

I do not understand what authors mean by bidirectional.

Response: Thanks for the question. The word “bidirectional” describes the changing behavior of the emissions of LIBRA at 980 and 1550 nm in response to temperature, which is explained in the section “Temperature response behavior and mechanism of LIBRA.” and illustrated in Fig. 3a in the manuscript. When LIBRA was dispersed in aqueous environment, 980 nm emission decreased while 1550 nm emission increased as temperature went up (Figure 3b). The tendency of temperature response for 980 and

1550 nm was in the reverse direction. Thus, we named such temperature response behavior of the luminescence intensities with the word “bidirectional”. Compared to one based on the single peak intensity, bidirectional response could effectively enhance the sensitivity of ratiometric nanothermometer (Fig. 3d). In the revised manuscript, the description in the section “Temperature response behavior and mechanism of LIBRA.” and Fig. 3a were amended for easy understanding of “bidirectional”.

“The change of the two emission bands responsive to temperature was in reverse direction, which was named as “bidirectional”, so that the ratio of emissions had larger change rate than the intensity of single emission to bring about a higher thermal sensitivity (Fig. 3a and 3d).”

Fig. 3 | **a** Schematic of bidirectional temperature response of LIBRA. **b** NIR luminescence spectra of LIBRA@PEG in aqueous dispersion at 283 to 363 K by external heating.

Fig. 3 | d Relative thermal sensitivity (S_r) of LIBRA@PEG with sensing strategy based on ratio of two peaks and single peak intensity. The temperature range is from 283 to 363 K.

Comments 8:

LIBRA with NaYF₄:50%Yb shell showed a 23-fold enhancement of the emission at 1550 nm compared to Core-LnNP (Fig. 2i). - the mechanisms of 1550 nm emission enhancement is not clearly explained in the manuscript. Having Yb doped shell should depopulate ⁴I_{11/2} level of Er, which is feeding ⁴I_{13/2} level. So, is the 23-fold enhancement absolute or relative to the 980 nm emission band?

Response: Thanks for the valuable questions. Firstly, the 23-fold enhancement of 1550 nm emission of LIBRA with NaYF₄:50% Yb shell was calculated with the comparison of the emission intensity at 1550 nm of core-only nanoparticle in aqueous environment. The NaYF₄:Yb shell layer could repair the surface defects of core nanoparticle to avoid dissipation of excitation energy by surface defects and circumvent the quenching effect from environment to the heavily doped Er³⁺ ions in the core nanoparticle, which is

explained in the section “Characterization of Lanthanide Ions doped nanocomposite as Bidirectional-responsive Ratiometric nanothermometer (LIBRA)”.

“The heavily doped Er^{3+} ions were favorable to the utilization of excitation energy for NIR luminescence and promoted the energy transfer to Yb^{3+} ions, but the direct luminescence quenching by solvent or ligand was the main concern of heavy doping design⁵⁶. Hence, luminescence spectroscopy was performed to see if the $\text{NaYF}_4:\text{Yb}$ shell layer on LIBRA could achieve the anti-quenching effect. NIR luminescence spectra of Core-LnNP and LIBRA dispersed in aqueous solution (both modified with DSPE-PEG for hydrophilization) revealed that the $\text{NaYF}_4:\text{Yb}$ shell of LIBRA substantially improved the NIR emissions of core nanoparticles. For example, LIBRA with $\text{NaYF}_4:50\%\text{Yb}$ shell showed a 23-fold enhancement of the emission at 1550 nm compared to Core-LnNP (Fig. 2i).”

Besides, as shown in Supplementary Fig. 17, the luminescence lifetime at $^4\text{I}_{11/2}$ level of Er^{3+} of core-active shell was obviously declined to $\sim 22 \mu\text{s}$ compared with the core-inert shell nanoparticle ($\sim 32 \mu\text{s}$). The declined lifetime of $^4\text{I}_{11/2}$ level of Er^{3+} ions might be caused by the enhanced quenching effect by aqueous environment through the energy migration among Yb^{3+} ions both in the core and shell. The shortened lifetime of $^4\text{I}_{11/2}$ level of Er^{3+} ions endowed by active shell could effectively suppress the upconversion process and facilitate a rapid nonradiative relaxation to $^4\text{I}_{13/2}$ level followed by 1550 nm emission so that 1550 nm luminescence intensity could be enhanced by $\text{NaYF}_4:\text{Yb}$ shell (Fig. 1).

Supplementary Fig. 17 Luminescence decay curve at 980 nm of nanoparticle with inert shell (NaErF₄:Yb@NaYF₄) and active shell (NaErF₄:Yb@NaYF₄:50%Yb) in aqueous environment.

Comments 9:

In general, often, no in-depth discussion on mechanism / phenomena behind is provided.

It is not enough to describe what one can see on the graphs. E.g. why the lifetimes (fig.4d) get shorter with temperature?

Response: Thanks for your suggestion. As shown in Fig. 4d, luminescence lifetime at 980 nm of NaErF₄:Yb@NaYF₄:Yb in DI-H₂O got shortened apparently compared to those dispersed in DMF. This means that water molecule has strong quenching effect to the energy level corresponding to 980 nm emission such as ⁴I_{11/2} of Er³⁺ and ²F_{5/2} of Yb³⁺ ions. According to the Raman spectra of water, intensity at 3400-3600 cm⁻¹ got stronger as temperature went up,¹² which means that quenching effect by water to the energy level of 980 nm emission got stronger accordingly. Thus, the luminescence lifetime at 980 nm was decreased by ~11 % from 15 to 55 °C (Supplementary Fig. 16).

Fig. 4 | d Luminescence decay curve at 980 and 1550 nm of NaErF₄:Yb@NaYF₄:Yb in the dispersion of DMF and H₂O under room temperature.

Supplementary Fig. 16 Luminescence lifetime at 980 nm and 1550 nm of NaErF₄:20%Yb@NaYF₄:Yb (LIBRA) in aqueous environment under varied temperature. Error bars, defined as s.d., were based on three measurements by spectrometry.

Comments 10:

Why both passive and active shells have similar relative sensitivities (Fig.4g) ?

Response: Thanks for the question. Nanoparticles with active shell exhibit higher relative sensitivity than that with inert (passive) shell. Yb^{3+} doping in the shell could facilitate the interaction of Er^{3+} ions and water molecule through energy migration among Yb^{3+} ions in the core and shell layer. As shown in Fig. 4g, relative sensitivity could be improved from ~ 1.6 to ~ 2 % K^{-1} at 283 K when active shell is used.

Fig. 4 | g Relative thermal sensitivity (S_r) of ratio (I_{1550}/I_{980}) of LIBRA@PEG with 0 and 50% Yb^{3+} in the shell.

Comments 11:

Why DMF quench I_{980} ?

Response: Thanks for the question. To further investigate the reason for the decrease of 980 nm emission intensity, LIBRA in solid state was degassed by nitrogen and then the luminescence spectra were measured under nitrogen atmosphere with varied temperature. As shown in Fig. 4b, 980 nm emission decreased while 1550 nm emission

slightly increased as temperature elevated. The weak thermal enhancement of I_{1550} could be attributed to the non-radiation transition of ${}^4I_{11/2} \rightarrow {}^4I_{13/2}$ because of multi-phonon relaxation (MPR).¹³ The decrement of 980 nm could be attributed to the thermal quenching of the nanocrystals. The thermally populated phonon relaxation could promote the non-radiative transition of excited energy level¹⁴. In DMF dispersion, the temperature response behavior of I_{980} of LIBRA was similar to the one observed in the solid state (Supplementary Fig. 9). Therefore, the quenching for 980 nm emission was not caused by DMF solvent but the intrinsic property of the nanoparticles. The corresponding discussion about this question is also organized and presented in the paragraph of “Temperature response behavior and mechanism of LIBRA.” of “Results” section in the revised manuscript.

Fig. 4 | b NIR luminescence spectra of LIBRA solid powder and those dispersed in water solution at 10, 50 and 90 °C.

Supplementary Fig. 9 NIR luminescence spectra of LIBRA dispersed in DMF at 10, 50 and 90 °C.

Comments 12:

What are the black up-arrows in Er (Fig.1) ?

Response: We appreciate this question. The black up-arrows in the energy level diagram of Er^{3+} in Fig. 1 are intended to indicate the upconversion process of Er^{3+} via excited state absorption by the 808 nm excitation or energy transfer from excited Er^{3+} ions. In the revised manuscript, the upconversion processes have been labeled and amended in Fig. 1. The upconversion processes are not mainly involved in this work so that Fig. 1 has been rearranged to be more focused on the energy migration among Yb^{3+} in both core and active shell and the interaction between $\text{Yb}^{3+}/\text{Er}^{3+}$ and water molecules.

Comments 13:

The intensity of 1550 nm emission increased gradually and the intensity at 980 nm decreased with temperature but what is the mechanism behind?

Response: Thanks for the question. The mechanism of intensity changes of 1550 and 980 nm emission is discussed in the section “Temperature response behavior and mechanism of LIBRA.” in the manuscript.

To decouple the contribution of Yb³⁺ and Er³⁺ in 980 nm emission and study the interaction between Er³⁺ ions and H₂O, NaErF₄:20%Y@NaYF₄ nanoparticles were synthesized with identical morphology and size as LIBRA (Supplementary Fig. 11) and the emission spectra were measured in aqueous environment under varied temperature (Supplementary Fig. 12). As temperature rose, 980 nm emission corresponding to ⁴I_{11/2}→⁴I_{15/2} transition of Er³⁺ decreased while 1550 nm emission corresponding to ⁴I_{13/2}→⁴I_{15/2} transition increased, which showed the same phenomenon as observed in LIBRA (NaErF₄:Yb³⁺@NaYF₄:Yb³⁺). As the stretch vibration at 3400-3600 cm⁻¹ of water molecule matches well with the energy gap between ⁴I_{11/2} and ⁴I_{13/2} states of Er³⁺ ions (Supplementary Fig. 14), the energy transfer between Er³⁺ ions and water molecules can occur (Coupling 1, ⁴I_{11/2} (Er³⁺) + ν = 0 (H₂O) → ⁴I_{13/2} (Er³⁺) + ν = 1 (H₂O)). The intensity of stretch vibration at 3400-3600 cm⁻¹ of water becomes stronger with temperature elevation¹², thus leading to the temperature responsive energy transfer process. As a consequence, the downshifting emission at 1550 nm increased with the rise of temperature. Meanwhile, the energy level of ⁴I_{11/2} was diminished, which was proved by the decrease of luminescence lifetime at 980 nm (Supplementary Fig. 13), and the corresponding 980 nm emission dropped as temperature.

Besides, O–H overtone transition ν = 0 → ν = 3 of water molecule peaking at ~10300 cm⁻¹, which matches well with ²F_{5/2} → ²F_{7/2} (~10200 cm⁻¹) transition of Yb³⁺

ions (Fig. 1), increases with temperature (Supplementary Fig. 15). Hence, a similar temperature responsive energy transfer process would also happen between Yb^{3+} and H_2O (Coupling 2, ${}^2\text{F}_{5/2}(\text{Er}^{3+}) + \nu = 0(\text{H}_2\text{O}) \rightarrow {}^2\text{F}_{7/2}(\text{Er}^{3+}) + \nu = 3(\text{H}_2\text{O})$), which was another possible reason for the decrease of 980 nm emission as temperature. Hence, LIBRA with Yb^{3+} doping in the shell give rise to higher relative sensitivity compared to the one without Yb^{3+} doping in the shell (Fig. 4g). Based on the suggestions of referee and the results of our work, the temperature sensing mechanism of LIBRA is summarized and proposed to be environment quenching assisted downshifting (EQAD).

The corresponding amendments are presented in the revised manuscript.

Supplementary Fig. 11 (a) TEM images of $\text{NaErF}_4:20\% \text{Y}$, $\text{NaErF}_4:20\% \text{Y}@\text{NaYF}_4$.

(b) Size distribution diagrams of $\text{NaErF}_4:20\% \text{Y}$, $\text{NaErF}_4:20\% \text{Y}@\text{NaYF}_4$. (c) Statistic

diameters and shell thicknesses of $\text{NaErF}_4:20\% \text{Y}$, $\text{NaErF}_4:20\% \text{Y}@\text{NaYF}_4$.

Supplementary Fig. 12 Emission spectra of NaErF₄:20% Y@NaYF₄ in the aqueous environment under varied temperatures, excited by 808 nm laser.

Supplementary Fig. 13 Luminescence lifetime at (a) 980 nm and (b) 1550 nm of NaErF₄:20% Y@NaYF₄ in aqueous dispersion at different temperatures. Error bars, defined as s.d., were based on three measurements by spectrometry.

Supplementary Fig. 14 Scheme of thermal response of NaErF₄:Y@NaYF₄ in aqueous environment.

Supplementary Fig. 15 Water absorption spectra from 9600 to 11000 cm⁻¹ under varied temperature.

Comments 14:

Why was a linear model proposed to model the experimental data set (see ref.12]) ?

Response: As shown in Figure 1, the interaction between Er³⁺ ion and water molecule, which can be described as $4I_{11/2}(\text{Er}^{3+}) + v = 0(\text{H}_2\text{O}) \rightarrow 4I_{13/2}(\text{Er}^{3+}) + v = 1(\text{H}_2\text{O})$ (Coupling 1 in the manuscript), could influence luminescence intensity at 980 and 1550

nm through with quenching effect of stretching vibration of water. Since Raman signal at 3400-3600 cm^{-1} of water (indicating the stretching vibration) increased homogeneously as temperature,¹² so the decrement of 980 nm and increment of 1550 nm changed as a linear function as temperature (Supplementary Fig. 2). Therefore, the ratio of I_{1550}/I_{980} as temperature is demonstrated in a linear model.

Supplementary Fig. 2 Normalized intensity and linear fit of 980 nm and 1550 nm of LIBRA under varied temperature. Error bars, defined as s.d., were based on three measurements by spectrometry.

Comments 15:

Why the emission lifetime of ${}^4I_{13/2}$ is so short? I would expect it to be significantly longer.

Response: The relatively short lifetime of ${}^4I_{13/2}$ is caused by several reasons. Firstly, the high doping concentration of Er^{3+} ions in the core of LIBRA leading to the enhanced cross-relaxation between Er^{3+} ions can significantly shorten the lifetime of ${}^4I_{13/2}$, of which similar phenomenon is reported by Meijerink *et al.*¹⁵. Secondly, the Yb^{3+} doped

thin shell layer of LIBRA builds an energy transfer pathway between Er^{3+} ions and water molecules for temperature response, which also results in the short lifetime of ${}^4\text{I}_{13/2}$. As reported by Fan Zhang *et al.*,¹⁶ both doping concentration of Er^{3+} in the core and the thickness of energy relay layer (shell layer with energy transmitter ions such as Yb^{3+}) would influence the emission lifetime of ${}^4\text{I}_{13/2}$ of Er^{3+} ions. On one hand, increasing the thickness of the energy relay layer from 0 to 7 nm led to a longer lifetime. On the other hand, at a defined thickness of the energy relay layer, the elevation of Er^{3+} concentration shortens the emission lifetime. For example, $\text{NaGdF}_4@\text{NaGdF}_4:\text{Yb}$, $\text{Er}@\text{NaYF}_4:\text{Yb}@\text{NaNdF}_4:\text{Yb}$ reported by Zhang *et al.* could be seen as a reference. With 45 % of Er^{3+} concentration in the core and without $\text{NaYF}_4:\text{Yb}$ energy relay layer, luminescence lifetime at 1525 nm of such nanoparticle was as short as 5.8 μs , which was pretty short compared to the millisecond-scale lifetime reported elsewhere. The LIBRA nanoparticles we synthesized have a high doping concentration of Er^{3+} ions (80 %) in the core and the $\text{NaYF}_4:\text{Yb}$ thin shell with the thickness of 1.45 nm was coated on the core, which led to the short emission lifetime of ${}^4\text{I}_{13/2}$.

Comments 16:

Some comments to figures: There is no error estimation of e.g. Fig.3, taking into account errors in establishing Er and Yb emission.

Response: Thanks for the valuable advice. The error estimations in Fig. 3e and 3f are supplemented in the revised manuscript, which are also shown below.

Fig 3 | **e** Repeatability of luminescence intensity ratio (LIR) with 4 cycles of heating and cooling between 308 and 323 K. Error bars, defined as s.d., were based on three measurements by spectrometry. **f** Ratio of emissions at 1550 and 980 nm (I_{1550}/I_{980}) under varied excitation power densities at room temperature. Error bars, defined as s.d., were based on three measurements by spectrometry.

Comments 17:

Why there is no temperature quenching at 1550 in DMF (fig.4b) when slight Yb quenching is observed in DMF?

Response: Firstly, to figure out the temperature response of NaErF₄:Yb@NaYF₄:Yb nanoparticle itself, we carried out the measurement of emission spectra in the solid powder state of NaErF₄:Yb@NaYF₄:Yb without any solvent. As shown in Fig. 4b and 4c, the intensity at 980 nm decreased while 1550 nm emission weakly increased as temperature elevated. The decrement of 980 nm could be attributed to the thermal quenching of the nanocrystals. The thermally populated phonon relaxation could promote the non-radiative transition of excited energy level.¹⁴ The small increment of 1550 nm may be caused by the multi-phonon relaxation (MPR) from ⁴I_{11/2} to ⁴I_{13/2}.^{13, 14}

In DMF dispersion, the thermal response behavior of 980 nm emission of LIBRA was well in line with the solid powder (Supplementary Fig. 8), indicating that DMF only showed weak quenching effect to the luminescence of LIBRA. DMF has a relatively weak absorbance peak near 1500 nm (Supplementary Fig. 9), of which the edge has a moderate overlap with the 1550 nm emission of LIBRA. In this case, the small increment of 1550 nm emission was reduced so that it kept almost unchanged in DMF.

The corresponding discussion about this question is also organized and presented in the paragraph of “Temperature response behavior and mechanism of LIBRA.” of “Results” section in the revised manuscript.

Supplementary Fig. 10 Absorption spectrum of DMF in the near infrared second window.

Comments 18:

Fig.6. Shouldn't LPS (and sepsis) induce fever (hyperthermia) rather than hypothermia?

Response: Thanks for the question. LPS was usually used to induce local inflammation, which can occur with fever (hyperthermia).¹⁷ However, high dose of LPS (50 mg kg⁻¹) injected intraperitoneally will induce sepsis (severe systemic inflammation) that can

cause whole body hypothermia (including brain) due to the endotoxin shock affecting the thermoregulatory of the body .¹⁸⁻²⁰ The temperature of the scalp was recorded by thermal camera (Supplementary Fig. 30). Meanwhile, we implanted a thermocouple into the brain of mouse and find that the tendency of temperature was similar to that by thermal camera (Fig. 6d and Supplementary Fig. 29). Three biologically independent mice were carried on the mentioned experiments above. Thus, a hypothermia model was successfully established by intraperitoneal injection of LPS. In the revised manuscript, the method of the establishment of sepsis induced hypothermia model in mouse is supplemented in the “Method” section.

“Establishment of sepsis induced hypothermia model in mouse.

To establish sepsis induced hypothermia mouse model, female Balb/c mice (n = 3) were intraperitoneally administrated with lipopolysaccharide (LPS) at a dose of 50 mg kg⁻¹, which will induce sepsis (severe systemic inflammation) that can cause whole body hypothermia (including brain) due to the endotoxin shock affecting the thermoregulatory of the body⁷¹⁻⁷³. After LPS injection, thermal camera was used to track the temperature of scalp and thermocouples implanted in the brain tissue and fixed on the abdomen skin were used to monitor the temperature changes of the brain and the body, respectively.”

Supplementary Fig. 30 (a-g) Time-course thermal images before and during LPS-induced hypothermia. (h) Corresponding quantitative temperature variations from thermal images, error bars were defined as s.d.

[FIGURE REDACTED]

Supplementary Fig. 29 Timeline and scheme of intracerebral temperature detecting through thermocouple before and during LPS-induced hypothermia.

Fig. 6 | d Intracerebral temperature variations recorded by thermocouple before and during LPS-induced hypothermia model. Error bars, defined as s.d., were based on three biologically independent mice.

Comments 20:

Can the NP penetrate through blood-brain barrier after I.V. injection?

Response: Thanks for the question. In this work, LIBRA nanoparticles are not likely and not necessary to penetrate through blood-brain barrier. As shown in Fig. 5a, no obvious luminescence signal outside brain blood vessel can be found in both control and hypothermia group.

Comments 21:

While the 808 nm is good for bio, the 980 nm emission from Yb overlap with water, thus on the path between the nanothermometers to detector, the amount/length/type of tissue may disrupt the 1550/980 ratio and may provide misleading data based on calibration data. Taking into account that no one knows the depth the emission comes from the calibration, the temperatures presented require more in-depth critical analysis and discussion.

Response: We appreciate the referee for the advice. In this work, the blood vessels observed in this work (inferior cerebral veins (ICV), the superior sagittal sinus (SSS), superficial veins (SV), and the transverse sinus (TS)) are vein vessels distributing on the top of cortex. The biological tissues including scalp, periosteum, skull and etc.

covering the veins have identical thicknesses, which are measured in this work to be 0.69 ± 0.02 mm, so it is reasonable to conclude that these veins have similar tissue depth and the condition of extinction coefficient. Hence, we established the setup simulating the condition of cerebral vessel in mouse for calibration curve measurement (Supplementary Fig. 31). In the setup for calibration curve measurement, the blood from vein is mixed with LIBRA and is filled into the capillary tube with diameter of 200 μm to simulate the cerebral blood vessels observed in the in vivo demonstration. The intact tissues above mouse brain including scalp, periosteum, skull and etc. are used to cover the capillary tube to build the tissue scattering and absorption parameters. The calibration curve obtained from the setup was shown in Supplementary Fig. 33.

For in vivo demonstration, normal and hypothermia mouse injected with LIBRA exhibited different luminescence intensity ratio in the cerebral blood vessels based on the near infrared luminescence imaging, which indicates that LIBRA can discriminate the temperature difference in the living body for hypothermia diagnosis (Fig. 6e and 6f). By using the calibration curve for in vivo thermometry to calculate the temperature difference of the cerebral blood vessels between normal and hypothermia mouse, it showed that the temperature of cerebral blood vessels obtained from the luminescence intensity ratio of LIBRA had a decrease of 3.94 $^{\circ}\text{C}$. These results imply that the relative temperature change detected by LIBRA can reflect the hypothermia symptom in cerebral vascular system and thus explains one of the reasons for the temperature change of the brain tissue. The fast diagnosis of hypothermia using luminescence intensity ratio thermometry of LIBRA is important to guide the treatment in time and

the high spatial resolution given by the NIR-II/III emission of LIBRA will also provide clear information of the hypothermia affected brain regions to evaluate the disease prognosis, which are the advances in the field of biological thermometry.

Reviewer #3 (Remarks to the Author):

Comments:

In this work, the authors proposed a bidirectional-responsive ratiometric nanothermometer (abbreviated as LIBRA) based on NaErF₄:Yb@NaYF₄:Yb nanoprobe for temperature imaging in vivo. The luminescence resonant energy transfer between lanthanide ions and water molecules resulted in temperature-sensitive emissions at 1550 nm and 980 nm, the emission intensity ratio of which can be applied for temperature detection. Based on the NaErF₄:Yb@NaYF₄:Yb nanoprobe, noninvasive temperature imaging of mouse cerebrovascular system was realized with a thermometric imaging resolution of ~200 μm. It is a systematic study and the topic itself may be of great interest for the researchers in this field. However, some points of the study should be further clarified before publication.

Comments 1:

1. The authors mentioned that few studies reported the temperature monitoring of micron-sized patterns in vivo because the luminescence intensity of NIR-II/III nanothermometer was not optimized. What is the NIR quantum yield of the proposed NaErF₄:Yb@NaYF₄:Yb nanoprobe?

Response: We appreciate this valuable advice. The relative quantum yield of LIBRA (QY_{LIBRA}) was determined by using IR-806 dye in DMF as a reference, which is reported to be 7.0 %²¹. The quantum yield was measured based on the method reported previously^{22, 23},

$$QY_{LIBRA} = QY_{IR-806} \cdot \frac{E_{LIBRA}}{E_{IR-806}} \cdot \frac{A_{IR-806}}{A_{LIBRA}} \cdot \frac{I_{IR-806}}{I_{LIBRA}} \cdot \left(\frac{N_{LIBRA}}{N_{IR-806}}\right)^2$$

where QY_{LIBRA} and QY_{IR-806} are the quantum yield of the LIBRA and IR-806, E_{LIBRA} and E_{IR-806} are the integrated emission intensity of LIBRA and IR-806, A_{LIBRA} and A_{IR-806} are the absorption of LIBRA and IR-806 in the dispersion, I_{LIBRA} and I_{IR-806} are the relative excitation intensity of LIBRA and IR-806 and N_{LIBRA} and N_{IR-806} are the refractive index of the solvents for dispersing the samples. The quantum yield of LIBRA was 2.12 ± 0.08 %.

Comments 2:

Supplementary Figure 4 shows the DLS measurements of DSPE-PEG modified LIBRA in deionized water, PBS and 37 °C FBS solution, which indicates the hydrodynamic diameters of DSPE-PEG modified LIBRA are much larger than that of the OA-capped LIBRA. Transmission electron microscopy images of the DSPE-PEG modified LIBRA in different solutions are suggested to be provided.

Response: We appreciate this valuable advice. TEM images of DSPE-PEG modified LIBRA in deionized water (DI-H₂O), PBS and 37 °C FBS solution were obtained and were presented in the revised manuscript as Supplementary Fig. 5. No apparent aggregate or dense cluster of nanoparticles was observed in the TEM samples of

different dispersions, which indicated good dispersity of nanoparticles in various aqueous environment. According to the TEM images, DSPE-PEG modified LIBRA in DI-H₂O exhibited similar distribution status as in cyclohexane after solvent volatilization because there is no other solute in the solution and the contrasts of OA and DSPE-PEG were too low to be imaged by TEM due to their small molecular weights. In PBS group, phosphates around the LIBRA separated out gradually as water in PBS volatilized and generated contrast so that thin coverings could be seen in Supplementary Fig. 5. Due to the complicated components in FBS solution such as proteins, electrolytes, carbohydrates, et al., which formed a relatively thick covering on the TEM copper grid and impeded the transmission of electrons, the TEM image of LIBRA in FBS solution was thus blurred, but the dispersity of the nanoparticles was still good (Supplementary Fig. 5).

Dynamic light scattering (DLS) measurement is based on the light scattering effect of particles to reflect the size distribution profile of particles in suspension. In aqueous environment, the nanocrystal, DSPE-PEG polymer and other solutes absorbed on the surfaces of nanoparticles can lead to the light scattering and contributed to hydrodynamic diameters so that the diameters in aqueous solutions were larger than the one provided by TEM, which only shows the contrast of the inorganic nanocrystals.

Supplementary Fig. 5. Dispersivity of LIBRA in (a) deionized water (DI-H₂O), (b) 1× PBS and (c) 37 °C FBS solution. Photographs (upper), TEM images (middle) and DLS measurements (lower) of LIBRA in DI-H₂O, PBS and 37 °C FBS solution. Scale bar in TEM images is 50 nm.

Comments 3:

Figure 3d displays the relative thermal sensitivity of LIBRA@PEG. The temperature uncertainty (δT) as another important thermal parameter is suggested to be provided.

Response: Thanks for the precious advice. The temperature uncertainties (δT) of LIBRA@PEG is calculated and supplemented in the revise manuscript. According to the equation 2,

$$\Delta T_{\min} = \frac{\sigma}{S_a} \quad (2)$$

where σ is the standard deviation of luminescence intensity ratio and S_a is the absolute sensitivity that equals to the slope of the fitting curve in the plot of luminescence intensity ratio *versus* temperature (Fig. 3c). The uncertainties were calculated and shown in Supplementary Fig. 7. Benefited from the ideal photostability of lanthanide ions doped nanoparticle, the temperature uncertainty by LIBRA was in low level (0.005 to 0.102 K at the range of 283 to 363 K).

In revised manuscript, related description is added in the “Results” section:

“The temperature uncertainties (δT) of LIBRA@PEG is defined according to the equation 2,

$$\Delta T_{\min} = \frac{\sigma}{S_a} \quad (2)$$

where σ is the standard deviation of luminescence intensity ratio and S_a is the absolute sensitivity that equals to the slope of the fitting curve in the plot of luminescence intensity ratio *versus* temperature (Fig. 3c). The δT of LIBRA@PEG were in low level, which were calculated to be 0.05 to 0.105 K at the range of 283 to 363 K (Supplementary Fig. 7).”

Supplementary Fig. 7. Temperature uncertainty of LIBRA under different temperatures.

Comments 4:

When LIBRA was dispersed in DMF, the intensity of 980-nm emission decreased while the 1550-nm emission was almost unchanged with increasing temperature.

Why? Please explain it.

Response: We appreciate this question. To better explain the temperature response behavior of LIBRA in DMF, the intrinsic temperature response of LIBRA was investigated first by detecting the emission spectra of LIBRA in the solid powder state without any solvent under nitrogen atmosphere with various temperatures. As shown in Fig. 4b and 4c, the intensity of 980 nm emission exhibited a mild decrease from 10 to 90 °C and the 1550 nm emission increased weakly as temperature elevated. The decrement of 980 nm could be attributed to the thermal quenching of the nanocrystals. The thermally populated phonon relaxation could promote the non-radiative transition of excited energy level¹⁴. The small increment of 1550 nm can be caused by the multi-phonon relaxation (MPR) from $^4I_{11/2}$ to $^4I_{13/2}$ ¹³.

When LIBRA was dispersed in DMF, the thermal response behavior of 980 nm emission was identical to the case in solid state (Supplementary Fig. 9), which means the decrease of 980 nm emission is the intrinsic thermal property of LIBRA rather than the effect of DMF. Meanwhile, the 1550 nm emission was nearly unchanged with temperature (Supplementary Fig. 9). DMF has a relatively weak absorbance peak near

1500 nm (Supplementary Fig. 10), of which the edge has a moderate overlap with the 1550 nm emission of LIBRA. In this case, the small increment of 1550 nm emission as observed in the solid state was further reduced in DMF so that the 1550 nm emission is nearly unchanged or exhibits very slight change with increasing temperature.

The corresponding discussion about this question is also organized and presented in the paragraph of “Temperature response behavior and mechanism of LIBRA.” of “Results” section in the revised manuscript.

Reference

1. Jia M, *et al.* Lanthanide-based ratiometric luminescence nanothermometry. *Nano Res.* **16**, 2949-2967 (2023).
2. Quintanilla M, Henriksen-Lacey M, Renero-Lecuna C, Liz-Marzán LM. Challenges for optical nanothermometry in biological environments. *Chem. Soc. Rev.* **51**, 4223-4242 (2022).
3. Bradac C, Lim SF, Chang HC, Aharonovich I. Optical nanoscale thermometry: from fundamental mechanisms to emerging practical applications. *Adv. Opt. Mater.* **8**, 2000183 (2020).
4. Bednarkiewicz A, Drabik J, Trejgis K, Jaque D, Ximendes E, Marciniak L. Luminescence based temperature bio-imaging: Status, challenges, and perspectives. *Appl. Phys. Rev.* **8**, 011317 (2021).
5. Nexha A, Carvajal JJ, Pujol MC, Díaz F, Aguiló M. Lanthanide doped luminescence nanothermometers in the biological windows: strategies and applications. *Nanoscale* **13**, 7913-7987 (2021).
6. Ansari AA, Parchur AK, Nazeeruddin MK, Tavakoli MM. Luminescent lanthanide nanocomposites in thermometry: Chemistry of dopant ions and host matrices. *Coord. Chem. Rev.* **444**, 214040 (2021).
7. Sun Y, Feng W, Yang P, Huang C, Li F. The biosafety of lanthanide upconversion nanomaterials. *Chem. Soc. Rev.* **44**, 1509-1525 (2015).

8. Gnach A, Lipinski T, Bednarkiewicz A, Rybka J, Capobianco JA. Upconverting nanoparticles: assessing the toxicity. *Chem. Soc. Rev.* **44**, 1561-1584 (2015).
9. Shen Y, *et al.* Reliable and Remote Monitoring of Absolute Temperature during Liver Inflammation via Luminescence-Lifetime-Based Nanothermometry. *Adv. Mater.* **34**, 2107764 (2022).
10. Qiu X, Zhou Q, Zhu X, Wu Z, Feng W, Li F. Ratiometric upconversion nanothermometry with dual emission at the same wavelength decoded via a time-resolved technique. *Nat. Commun.* **11**, 4 (2020).
11. del Rosal B, *et al.* Infrared-Emitting QDs for Thermal Therapy with Real-Time Subcutaneous Temperature Feedback. *Adv. Funct. Mater.* **26**, 6060-6068 (2016).
12. Nilsson A, Pettersson LGM. The structural origin of anomalous properties of liquid water. *Nat. Commun.* **6**, 8998 (2015).
13. Jia M, *et al.* NIR - II/III Luminescence Ratiometric Nanothermometry with Phonon - Tuned Sensitivity. *Adv. Opt. Mater.* **8**, 1901173 (2020).
14. Wang Y, Chen B, Wang F. Overcoming thermal quenching in upconversion nanoparticles. *Nanoscale* **13**, 3454-3462 (2021).
15. Wang Z, Meijerink A. Concentration Quenching in Upconversion Nanocrystals. *J. Phys. Chem. C* **122**, 26298-26306 (2018).
16. Fan Y, *et al.* Lifetime-engineered NIR-II nanoparticles unlock multiplexed in vivo imaging. *Nat. Nanotechnol.* **13**, 941-946 (2018).
17. Xu M, *et al.* Ratiometric nanothermometer in vivo based on triplet sensitized upconversion. *Nat. Commun.* **9**, 2698 (2018).
18. Romanovsky AA, Shido O, Sakurada S, Sugimoto N, Nagasaka T. Endotoxin shock: thermoregulatory mechanisms. *Am. J. Physiol.* **270**, R693-703 (1996).
19. Wiewel MA, *et al.* Risk factors, host response and outcome of hypothermic sepsis. *Critical Care* **20**, 328 (2016).
20. Blanqué R, Meakin C, Millet S, Gardner CR. Hypothermia as an indicator of the acute effects of lipopolysaccharides: comparison with serum levels of IL1 beta, IL6 and TNF alpha. *Gen. Pharmacol.* **27**, 973-977 (1996).
21. Bao G, *et al.* Enhancing Hybrid Upconversion Nanosystems via Synergistic Effects of Moiety Engineered NIR Dyes. *Nano Lett.* **21**, 9862-9868 (2021).

22. Williams ATR, Winfield SA, Miller JN. Relative fluorescence quantum yields using a computer-controlled luminescence spectrometer. *Analyst* **108**, 1067-1071 (1983).
23. Chen G, *et al.* Core/Shell NaGdF₄:Nd³⁺/NaGdF₄ Nanocrystals with Efficient Near-Infrared to Near-Infrared Downconversion Photoluminescence for Bioimaging Applications. *ACS Nano* **6**, 2969-2977 (2012).

REVIEWER COMMENTS

Reviewer #1 (Remarks to the Author):

In this revised version authors have modified their manuscript following all the comments raised previously. Regarding my previous comments, their answer justifying why they were not using lifetime for sensing temperature. It is very clear that for fast changes of temperature, lifetime is not valid.

I have more problems with the answer from authors regarding how trustable is the thermal readout providing by their nanoparticles. The key point here are the data included in Figure 32 and 33 of response letter. The calibration curve depends on the medium in a very strong manner. Indeed, if authors combine two figures into one, the calibration obtained in conditions simulating in vivo is not parallel to any of the calibrations obtained in liquids (including blood). Also note that what they call in vivo conditions could differ a lot from the response of their nanothermometers in vivo. For instance, in their experiments there is not special care to control humidity of the tissues so the tissue properties during calibration can be completely different than those of tissues in live animals. Also, from the data in the text Figure 6e the ratio in the vessels vary from 2 down to 0.6. According to Figure 33, these intensity ratios lead to values of temperature that make no sense. A value of intensity ratio of 0.6 corresponds to a temperature well below 20°C. In my opinion, in the best of the cases authors can measure only changes in relative temperature. But even in this scenario I am afraid that the temperature they are measuring is not trustable.

I must recognize that the idea of the manuscript is very nice and that authors are a reference in the field of in vivo imaging and sensing with nanoparticles. But it seems that current approach is not valid to provide absolute and trustable thermal images at in vivo. I am sorry to say that unless authors can provide solid data on the reliability of their thermal probes at the in vivo level I can not recommend publication of this article.

Reviewer #2 (Remarks to the Author):

I have no further questions and suggest acceptance of the paper, which has been significantly improved and comments to all reviewers have been satisfactory answered.

Reviewer #3 (Remarks to the Author):

The article, in its present form, is much improved with respect to its previous edition. All the questions I raised have been well addressed or explained. I recommend it for publication without further revision.

Responses to the Reviewers' Reports Manuscript (NCOMMS-22-37056A)

Reviewer #1 (Remarks to the Author):

In this revised version authors have modified their manuscript following all the comments raised previously. Regarding my previous comments, their answer justifying why they were not using lifetime for sensing temperature. It is very clear that for fast changes of temperature, lifetime is not valid.

I have more problems with the answer from authors regarding how trustable is the thermal readout providing by their nanoparticles. The key point here are the data included in Figure 32 and 33 of response letter. The calibration curve depends on the medium in a very strong manner. Indeed, if authors combine two figures into one, the calibration obtained in conditions simulating in vivo is not parallel to any of the calibrations obtained in liquids (including blood). Also note that what they call in vivo conditions could differ a lot from the response of their nanothermometers in vivo. For instance, in their experiments there is not special care to control humidity of the tissues so the tissue properties during calibration can be completely different than those of tissues in live animals. Also, from the data in the text Figure 6e the ratio in the vessels vary from 2 down to 0.6. According to Figure 33, these intensity ratios lead to values of temperature that make no sense. A value of intensity ratio of 0.6 corresponds to a temperature well below 20°C. In my opinion, in the best of the cases authors can measure only changes in relative temperature. But even in this scenario I am afraid that the temperature they are measuring is not trustable. I must recognize that the idea of the manuscript is very nice and that authors are a reference in the field of in vivo imaging and sensing with nanoparticles. But it seems that current approach is not valid to provide absolute and trustable thermal images at in vivo. I am sorry to say that unless authors can provide solid data on the reliability of their thermal probes at the in vivo level I can not recommend publication of this article.

Response: We appreciate the reviewer for the comment to improve this work. To circumvent the possible difference of the conditions between the bio-tissues *ex vivo* and

in vivo, we employed the photon diffusion theory and established the light propagation model to compute the luminescence intensities and their corresponding ratio after light propagation in the medium with absorption and scattering.

Diffuse light propagation in the medium with scattering and absorption can be predicted by using a photon diffusion equation (*Inverse Probl.* 15, R41 (1999)) given as,

$$\frac{\partial \Phi(\vec{r}, t)}{c \partial t} + \mu_a \Phi(\vec{r}, t) - \nabla \cdot [D \nabla \Phi(\vec{r}, t)] = S(\vec{r}, t) \quad (1)$$

where $\Phi(\vec{r}, t)$ denotes fluence rate (or intensity) of light at location \vec{r} and time point t . The medium is characterized by its absorption coefficient (μ_a) and reduced scattering coefficient (μ'_s). On the right side of the equation, $S(\vec{r}, t)$ is the source term. Furthermore, D is referred to as the diffusion coefficient, which can be obtained from,

$$D = \frac{1}{3(\mu_a + \mu'_s)} \quad (2)$$

For a time-independent point source, $S(\vec{r}, t) = \delta(\vec{r})$, the solution to Eq. (1) is typically known as the Green's function,

$$\Phi(\vec{r}) = \frac{1}{4\pi D r} e^{-\mu_{eff} \cdot \vec{r}} \quad (3)$$

where μ_{eff} denotes the effective attenuation coefficient that is expressed as,

$$\mu_{eff} = \sqrt{\frac{\mu_a}{D}} = \sqrt{3\mu_a(\mu_a + \mu'_s)} \quad (4)$$

In the case of this work, the luminescence propagation of the LIBRA nanoparticles is described as follows. As shown in Supplementary Fig. 33, the nanothermometer (LIBRA) as luminescence source is embedded in the biological tissue (medium). The incident excitation propagates in the medium and irradiates the luminescence source in the medium at a depth of d . The luminescence after propagation is measured on the top surface of the medium. The solution of LIR-temperature correlation is divided into three stages: 1) solving for the fluence intensity of incident excitation in the medium, 2) solving for the luminescence intensity of the luminescence source propagating in the medium, and 3) establishing the relationship of LIR and temperature in the medium.

Supplementary Fig. 33 Schematic of excitation and luminescence propagations in a medium with scattering and absorption. Incident excitation with a fluence intensity, $\Phi_{ex}(\lambda_{ex}, d)$, excites the luminescence source (green) embedded at a depth of d inside the medium to generate the local luminescence intensity, $LLI(\lambda_{em}, T)$. The luminescence propagates in the medium and is measured on the top surface of the medium to be $I(\lambda_{em})$.

For incident excitation with initial power I_0 at wavelength λ_{ex} , the Green's function solution of diffusion equation (Eq. 1) can be used to determine the excitation light field $\Phi_{ex}(\lambda_{ex}, d)$ at depth d ,

$$\Phi_{ex}(\lambda_{ex}, d) = I_0 \cdot \frac{1}{4\pi D(\lambda_{ex}) \cdot d} e^{-\mu_{eff}(\lambda_{ex})d} \quad (5)$$

Afterwards, local luminescence intensity (LLI), which is the original intensity emitted by the luminescence source, is calculated. This is the result of excitation field, the absorption of luminescence source and its quantum yield. At a given temperature (T), LLI of the emission at wavelength λ_{em} is as follows:

$$LLI(\lambda_{em}, T) = \Phi_{ex}(\lambda_{ex}, d) \cdot \mu_{a,l}(\lambda_{ex}) \cdot \sigma(\lambda_{em}, T) \quad (6)$$

where $\Phi_{ex}(\lambda_{ex}, d)$ is the excitation field, $\mu_{a,l}(\lambda_{ex})$ is the absorption of the luminescence source, and $\sigma(\lambda_{em}, T)$ is its corresponding quantum yield.

For the ratiometric thermometry in this work, the temperature is correlated with the ratio of LLI of the emissions at two different wavelengths (λ_{em1} and λ_{em2}), which is also equal to the ratio of quantum yields (σ) of the two emissions. The linear relationship between the ratio of LLI and temperature (T) can be written as the following form,

$$T = \alpha \cdot \frac{\sigma(\lambda_{em2}, T)}{\sigma(\lambda_{em1}, T)} + \beta \quad (7)$$

where α and β are constant, λ_{em1} and λ_{em2} in this work are the emissions at 980 and 1550 nm, respectively.

The luminescence intensity of the luminescence source after light propagation in the medium, $I(\lambda_{em})$, is obtained in the experiment by bioimaging. The Green's function is used to describe $I(\lambda_{em})$ of emission at wavelength λ_{em} , which refers to $LLI(\lambda_{em}, T)$:

$$I(\lambda_{em}) = LLI(\lambda_{em}, T) \cdot \frac{1}{4\pi D(\lambda_{em}) \cdot d} e^{-\mu_{eff}(\lambda_{em})d} \quad (8)$$

By substituting $LLI(\lambda_{em}, T)$ in Eq. 8 with Eq. 5 and 6, $I(\lambda_{em})$ can be expressed as:

$$I(\lambda_{em}) = \frac{I_0}{16\pi^2 d^2 D(\lambda_{ex}) D(\lambda_{em})} e^{-(\mu_{eff}(\lambda_{ex}) + \mu_{eff}(\lambda_{em})) \cdot d} \cdot \mu_{a,l}(\lambda_{ex}) \cdot \sigma(\lambda_{em}, T) \quad (9)$$

After light propagation in the medium, the luminescence intensities of the emissions at two wavelengths, $I(\lambda_{em1})$ and $I(\lambda_{em2})$, can be described by Eq. 9 and are measured in the experiment. The ratio of $I(\lambda_{em1})$ and $I(\lambda_{em2})$ is denoted as R_I that is,

$$\begin{aligned} R_I &= \frac{I(\lambda_{em2})}{I(\lambda_{em1})} \\ &= \frac{\frac{I_0}{16\pi^2 d^2 D(\lambda_{ex}) D(\lambda_{em2})} e^{-(\mu_{eff}(\lambda_{ex}) + \mu_{eff}(\lambda_{em2})) \cdot d} \cdot \mu_{a,f}(\lambda_{ex}) \cdot \sigma(\lambda_{em2}, T)}{\frac{I_0}{16\pi^2 d^2 D(\lambda_{ex}) D(\lambda_{em1})} e^{-(\mu_{eff}(\lambda_{ex}) + \mu_{eff}(\lambda_{em1})) \cdot d} \cdot \mu_{a,f}(\lambda_{ex}) \cdot \sigma(\lambda_{em1}, T)} \\ &= \frac{D(\lambda_{em1})}{D(\lambda_{em2})} e^{(\mu_{eff}(\lambda_{em1}) - \mu_{eff}(\lambda_{em2})) \cdot d} \cdot \frac{\sigma(\lambda_{em2}, T)}{\sigma(\lambda_{em1}, T)} \end{aligned} \quad (10)$$

Based on Eq. 7, the ratio of quantum yields of λ_{em2} and λ_{em1} can be expressed as,

$$\frac{\sigma(\lambda_{em2}, T)}{\sigma(\lambda_{em1}, T)} = \frac{T - \beta}{\alpha} \quad (11)$$

A correction term, γ , is further defined that is given as:

$$\gamma = \frac{D(\lambda_{em1})}{D(\lambda_{em2})} e^{(\mu_{eff}(\lambda_{em1}) - \mu_{eff}(\lambda_{em2})) \cdot d} \quad (12)$$

By substituting the corresponding items with Eq. 11 and 12, Eq. 10 can be expressed as,

$$R_I = \frac{T - \beta}{\alpha} \cdot \gamma$$

or:

$$T = \frac{\alpha}{\gamma} R_I + \beta \quad (13)$$

The term γ is related to the absorption and scattering of the medium at certain depth, which is used to correct the correlation of temperature (T) and the ratio of the luminescence intensities ($I(\lambda_{em})$) detected in the experiment that are derived from the local luminescence intensity of LIBRA nanothermometers and propagate in the biological tissue. Thus, the accurate relationship between LIR and temperature can be established as Eq. 13.

Next, the validation of the derived Eq. 13 was carried out by measuring the LIR under a series of temperature using the medium with known absorption and scattering coefficients. For this purpose, polydimethylsiloxane (PDMS) phantoms (Phantom A, B, C, D) were prepared with determined and stable thicknesses and optical properties (Supplementary Fig. 34, Supplementary Table 3). The phantoms consist of polydimethylsiloxane (PDMS) elastomer as matrix and the μ_a and μ'_s of phantoms are adjusted by integrating different levels of carbon black powder and titanium dioxide, respectively. The optical properties at 980 nm and 1550 nm of the phantoms are calibrated using inverse adding-doubling (IAD) method (*Appl. Opt.* 32, 399-410 (1993)).

Supplementary Fig. 34 Images of Phantom A, B, C, D with different optical properties (absorption coefficient (μ_a) and reduced scattering coefficient (μ'_s)) and thicknesses.

Supplementary Table 3 Thicknesses and optical properties (absorption coefficient (μ_a) and reduced scattering coefficient (μ'_s)) of phantoms.

Phantom	A	B	C	D
Thickness (cm)	0.050	0.120	0.150	0.130
μ_a (cm^{-1}), 980 nm	1.075	0.615	0.496	0.479
μ'_s (cm^{-1}), 980 nm	6.133	5.662	5.584	7.956
μ_a (cm^{-1}), 1550 nm	2.729	2.430	2.372	2.239
μ'_s (cm^{-1}), 1550 nm	2.628	2.486	2.371	4.321

For the measurement of LIR-temperature relationship with the use of phantom, the experimental setup was depicted in Supplementary Fig. 35. A continuous-wave (CW) 808 nm laser coupled with a collimator and a diffuser was employed as excitation source. LIBRA dispersed in aqueous solution was added into a capillary tube and the phantom was covered on the top. A high-precision heater was used for controlling temperature of LIBRA dispersion within a range of 26-46 °C and a thermocouple was inserted in the aqueous dispersion for real-time temperature recording. NIR-II imaging system was used to record the images collected by 900 nm and 1400 nm long pass filters to calculate the LIR of the emission bands at 980 nm and 1550 nm. During data acquisition, the temperature of LIBRA dispersion was controlled continuously and the NIR-II/III images were acquired at a frame rate of 1 fps and the exposure time was set to 200 ms. The thermocouple recorded the temperature change simultaneously with a sampling interval of 1 s.

[FIGURE REDACTED]

Supplementary Fig. 35 Experimental setup for the measurement of temperature dependent luminescence intensity ratio with different phantoms based on NIR-II imaging.

The LIR of the emissions at 1550 and 980 nm (I_{1550}/I_{980}) acquired by the NIR-II/III imaging and the temperature detected simultaneously were plotted together. The relationship of LIR and temperature was also computed by using Eq. 13. The parameters of α and β in Eq. 13 were fixed and determined to be 15.5 and -66.5, respectively, based on the experimental measurements. The correction term γ for Phantom A to D were calculated according to the thicknesses and optical properties of different phantoms that were 0.71, 0.59, 0.54 and 0.53, respectively. The fitting curves for LIR and temperature relationship for different phantoms are presented as: $T = 21.8 R_I - 66.5$ (Phantom A); $T = 26.3 R_I - 66.5$ (Phantom B); $T = 28.7 R_I - 66.5$ (Phantom C); $T = 29.2 R_I - 66.5$ (Phantom D). The fitting curves matched well with the experimental results (correlation coefficient >0.990), which indicated that the established model can accurately describe the effects of medium with absorption and scattering on the luminescence signals, and the proposed equation to calibrate the LIR and temperature relationship is feasible. Given by the light extinction parameters of the biological tissues including skin, scalp, blood, etc. based on reported works (*Nat. Photonics* 8, 723–730 (2014); *J. Biomed. Opt.* 14, 034001 (2009)), the calibrated equation calculated as $T = 22.9 R_I - 66.5$ is used for the temperature detection of mice cerebral vessels *in vivo*.

Supplementary Fig. 36 Experimental measurements and fitting curves of the relationship of luminescence intensity ratio R_I (I_{1550}/I_{980}) and temperature in different phantoms.

Then, the validated equation for describing the temperature and LIR correlation was employed to measure the temperature of cerebral vessels in mice based on the imaging results obtained by the emissions at 980 and 1550 nm of LIBRA. Normal and hypothermia mice injected with LIBRA exhibited different temperatures based on luminescence intensity ratio in the cerebral blood vessels, which confirmed that LIBRA can discriminate the temperature difference in the living body for hypothermia diagnosis (Fig. 6e). The average temperatures of ICV, SSS and TS in hypothermia group were determined to be 37.1, 37.2 and 36.4 °C, respectively, and the ones in control group were 29.2, 28.7 and 28.3 °C, respectively, indicating that the temperature change of cerebral blood vessel for hypothermia model could be detected by LIBRA.

[PANEL A OF FIGURE REDACTED]

Fig. 6 | Cerebral vessel temperature detection using LIBRA in hypothermia mouse and control mouse. a Scheme of cerebral vessel temperature detection through LIBRA. Thermal images of mouse scalp detected by infrared thermal camera in **b** control and **c** hypothermia groups. **d** Temperature variations of brain tissue and abdomen skin recorded by thermocouple before and during LPS-induced hypothermia model. Error bars, defined as s.d., were based on three biologically independent mice. **e** Ratiometric temperature imaging of cerebral vessel using LIBRA for hypothermia (left) and control

group (right) mouse. **f** Statistical histograms of ratios for cerebrovascular of control and hypothermia mice. Error bars, defined as s.d., were based on three biologically independent mice.

Reviewer #2 (Remarks to the Author):

I have no further questions and suggest acceptance of the paper, which has been significantly improved and comments to all reviewers have been satisfactory answered.

Response: We appreciate the reviewer for the comment and approval of our work.

Reviewer #3 (Remarks to the Author):

The article, in its present form, is much improved with respect to its previous edition. All the questions I raised have been well addressed or explained. I recommend it for publication without further revision.

Response: We appreciate the reviewer for the approval of our work.

REVIEWERS' COMMENTS

Reviewer #1 (Remarks to the Author):

The response from authors is convincing. Manuscript ready for publication

Responses to the Reviewers' Reports Manuscript (NCOMMS-22-37056B)

Reviewer #1 (Remarks to the Author):

The response from authors is convincing. Manuscript ready for publication

Response: We appreciate the reviewer for the approval of this work.